# Multiple carbamylation events are required for differential modulation of Cx26 hemichannels and gap junctions by CO$_2$

Sarbjit Nijjar[1], Deborah Brotherton[1], Jack Butler[1], Valentin-Mihai Dospinescu[1], Harry G. Gannon[2], Victoria Linthwaite[2], Martin Cann[2], Alexander Cameron[1] and Nicholas Dale[1]

[1]*School of Life Sciences, University of Warwick, Coventry, UK*
[2]*Department of Biosciences, Durham University, Durham, UK*

Handling Editors: Kim Barrett & Rajini Rao

The peer review history is available in the Supporting Information section of this article (https://doi.org/10.1113/JP285885#support-information-section).

**Abstract figure legend** Cx26 hemichannels and gap junction channels are differentially modulated by two carbamylation events on K125 and K108. Hemichannels at low $P_{CO_2}$ are ordinarily closed. When $P_{CO_2}$ rises sufficiently, both K125 and K108 can become carbamylated. Only carbamylation of K125 (orange) is necessary and sufficient to open the hemichannel. By contrast, gap junction channels at low $P_{CO_2}$ are ordinarily open. When $P_{CO_2}$ rises, carbamylation of both K125 and K108 (both orange) is required for gap junction channel closure. The structures are based on structure 8QA2 from the Protein Data Bank.

**Sarbjit Nijjar** received a PhD in developmental biology from KCL. His research in the laboratory of Professor Nicholas Dale at Warwick University has focused on understanding the regulation of the membrane channel protein connexin 26, by carbon dioxide, and how this may impact physiological processes such as breathing and hearing. **Deborah Brotherton** spent 12 years in drug discovery for cancer and inflammation after receiving her PhD in structural biochemistry from the University of Cambridge. Moving to the Structural Genomics Consortium, Oxford, and then the University of Warwick, her work has centred on CO$_2$-responsive ion channels using structural and biophysical techniques. **Jack Butler** is completing his PhD in Life Sciences at the University of Warwick under the supervision of Professor Nicholas Dale. His current research has focused on understanding the regulation, by carbon dioxide, of the membrane channel protein connexin 32 and how this influences physiological processes such as myelin maintenance.

S. Nijjar, D. Brotherton, and J. Butler, contributed as Joint first authors.

**Abstract**  $CO_2$ directly modifies the gating of connexin26 (Cx26) gap junction channels and hemichannels. This gating depends upon Lys125, and the proposed mechanism involves carbamylation of Lys125 to allow formation of a salt bridge with Arg104 on the neighbouring subunit. We demonstrate via carbamate trapping and tandem mass spectrometry that five Lys residues within the cytoplasmic loop, including Lys125, are indeed carbamylated by $CO_2$. The cytoplasmic loop appears to provide a chemical microenvironment that facilitates carbamylation. Systematic mutation of these Lys residues to Arg shows that only carbamylation of Lys125 is essential for hemichannel opening. By contrast, carbamylation of Lys108 and Lys125 is essential for gap junction closure to $CO_2$. Chicken (*Gallus gallus*) Cx26 gap junction channels lack Lys108 and do not close to $CO_2$, as shown by both a dye transfer assay and a high-resolution cryogenic electron microscopy structure. The mutation Lys108Arg prevents $CO_2$-dependent gap junction channel closure in human Cx26. Our findings directly demonstrate carbamylation in connexins, provide further insight into the differential action of $CO_2$ on Cx26 hemichannels and gap junction channels, and increase support for the role of the N-terminus in gating the Cx26 channel.

(Received 31 October 2023; accepted after revision 13 January 2025; first published online 4 February 2025)

**Corresponding author** N. Dale: School of Life Sciences, University of Warwick, Coventry, CV4 7AL, UK.    Email: N.E.Dale@warwick.ac.uk

**Key points**

- Direct evidence of carbamylation of multiple lysine residues in the cytoplasmic loop of Cx26.
- Concentration-dependent carbamylation at lysines 108, 122 and 125.
- Only carbamylation of lysine 125 is essential for hemichannel opening to $CO_2$.
- Carbamylation of lysine 108 along with lysine 125 is essential for $CO_2$-dependent gap junction channel closure.

## Introduction

$CO_2$ is an almost universal by-product of metabolism. It is produced from oxidative phosphorylation and a range of decarboxylation reactions. In mammals, the resting partial pressure of $CO_2$ in arterial blood ($P_{CO_2}$) is around 40 mmHg. Lorimer speculated that, given this abundance of $CO_2$, it could carbamylate proteins via a spontaneous and labile covalent bond to a primary amine, such as those of lysine residues or the N-terminal amines of polypeptide chains (Lorimer, 1983). This has long been known to occur in RuBisCo (Lorimer & Miziorko, 1980; Lundqvist & Schneider, 1991), an important enzyme in photosynthetic carbon fixation, and haemoglobin, where it mediates the Bohr effect (Kilmartin & Rossi-Bernardi, 1971). More recently, mass spectrometry (MS) methods that trap these labile carbamates have shown that post-translational modification of proteins by carbamylation is relatively prevalent (Blake & Cann, 2022; King et al., 2022; Linthwaite et al., 2018).

We have shown that connexin 26 (Cx26) is directly sensitive to $CO_2$ (Brotherton et al., 2022; de Wolf et al., 2017; Dospinescu et al., 2019; Huckstepp et al., 2010a; Meigh et al., 2013). The hemichannels open to $CO_2$

and permit the efflux of ATP, which is important in controlling breathing (Huckstepp et al., 2010b; van de Wiel et al., 2020). Our data, supported by extensive mutational analysis, suggests that carbamylation of K125 and the formation of carbamate bridges to R104 of the neighbouring subunit are essential for opening the hemichannel in response to $CO_2$ (Dospinescu et al., 2019; Meigh et al., 2013; Meigh et al., 2015). By developing a dominant negative subunit (Cx26[K125R,R104A]) which can neither bind $CO_2$ nor form a carbamate bridge to a neighbouring wild-type (WT) subunit that has bound $CO_2$, we have shown that the binding of $CO_2$ to the carbamylation motif of Cx26 contributes to around half of the centrally generated regulation of breathing at modest levels of hypercapnia (van de Wiel et al., 2020).

Cx26, like most connexins, can form unopposed membrane channels, called hemichannels and gap junction channels. The latter are formed when two hemichannels in apposed cells dock to form an aqueous channel between cells (Stout et al., 2002; Stout et al., 2004). A single connexin subunit has four transmembrane helices. Cx26 hemichannels are hexameric structures with the six subunits arranged around a central pore (Maeda et al., 2009; Muller et al., 2002; Oshima et al., 2007). A

gap junction channel is thus a dodecameric complex formed from two hexamers docking together via specific interactions between extracellular residues (Maeda et al., 2009). The N-terminal cytoplasmic helix from each sub-unit, which points into the pore, is thought to regulate the opening and closing of both hemichannels and gap junctions. Although modest elevations of $P_{CO_2}$ opens hemichannels, they close gap junction channels (Nijjar et al., 2021). Strangely, this closing action also depends on the carbamylation of K125 (Nijjar et al., 2021). We have recently obtained cryogenic electron microscopy (cryo-EM) structures of the Cx26 gap junction channels at different levels of $P_{CO_2}$ at constant pH. Consistent with the $CO_2$-dependent closure of the gap junction channel, the N-terminal helices form more of a constriction at high $P_{CO_2}$ relative to low $P_{CO_2}$ (Brotherton et al., 2022).

For carbamylation to occur, the target primary amine cannot be protonated. Lysine side chains have a pKa of ~10 in physiological conditions, so they will normally be protonated. Therefore, within the structure of the protein, a 'carbamylatable' lysine must be in a micro-environment where the $pK_a$ of the $\varepsilon$-amine is substantially lowered (Blake & Cann, 2022). In effect, the three-dimensional protein structure and resulting chemical microenvironment determine which lysine residues may be carbamylated and thus imparts specificity to this post-translational modification mechanism.

We have now applied MS methods and carbamate trapping to directly demonstrate the carbamylation of Cx26 (Linthwaite et al., 2018). We find that several lysines within the cytoplasmic loop can be carbamylated. However, carbamylation of these additional lysines is not required for hemichannel opening to $CO_2$. In parallel, we have solved the structure of chicken Cx26 gap junction channels at high $P_{CO_2}$. We observe that the maps resemble the more open structure of human Cx26 gap junction channels obtained at low rather than high $P_{CO_2}$. Based on this observation, we investigated the effect of $CO_2$ on chicken Cx26 gap junction channels and find that they do not close in response to $CO_2$. This led us to uncover the necessity of carbamylation of K108, one of the candidates revealed by the MS screen, for gap junction closure. Our work develops an understanding of carbamylation as a mechanism of ion channel modulation by demonstrating that multiple carbamylation events are required for differential regulation of ion channel function.

## Methods

### Purification of human Cx26

Human connexin 26 protein (HsCx26) for MS and trapping experiments was expressed and purified with a thrombin-cleavable His(6) affinity tag on the C-terminus using baculovirus in *Spodoptera frugiperda* (Sf9) cells, as has been described previously (Brotherton et al., 2022). Following elution of the protein from the NiNTA resin dialysis of the affinity purified protein was carried out against a buffer containing 10 mM sodium phosphate pH 7.4, 500 mM NaCl, 5% glycerol, 1 mM dithiothreitol (DTT), 0.03% dodecyl-$\beta$-D-maltoside (DDM) (Glycon Biochemicals GMBH), to remove histidine, before flash-freezing the protein.

### CO₂ trapping

Purified hCx26 (0.7 mg) in stabilising solution (100 mM NaCl, 1.25 mM NaH$_2$PO$_4$, 3 mM KCl, 10 mM D-glucose, 1 mM MgSO$_4$, 2 mM CaCl$_2$, 0.03 % (w/v) $n$-Dodecyl $\beta$-D-maltoside, 1 mM DTT, pH 7.5) was incubated with NaHCO$_3$ (50 mM, equivalent to 90 mmHg $P_{CO_2}$ for hypercapnic conditions or 15 mM, equivalent to 20 mmHg $P_{CO_2}$ for hypocapnic conditions) and added to a pH stat (Titrando 902, Metrohm) equipped with pH probe and burette. Triethyloxonium tetrafluoroborate (TEO) (280 mg, Sigma UK) was added in three stepwise increments in phosphate buffer (50 mM, 1 ml, pH 7.4). The reaction pH was maintained at 7.4 with the addition of 1 M NaOH throughout, the pH was stabilised until reaction completion (60 min). The trapped protein was dialysed overnight (1 l, dH$_2$O, 4°C).

### Mass spectrometry

The post-dialysis reaction trapping sample supernatant was removed using vacuum centrifugation. The dried protein sample was resuspended in 8 M urea and reduced with DTT (25 mM final concentration) at 37°C for 1 h. The sample was then alkylated with iodoacetamide (40 mM final concentration) in the dark for 1 h. This sample was centrifuged at 1000 $g$ for 5 min, and the soluble super-natant was removed. The sample was diluted to 1 M urea and digested with trypsin gold (mass spectrometry grade, Promega) in a 1:25 (w/w) ratio overnight at 37°C. The digested solution was then desalted on a C18 column and analysed by electrospray ionisation (ESI)-MS/MS on an LTQ Orbitrap XL mass spectrometer (Thermo) coupled to an Ultimate 3000 nano-high-performance liquid chromatography instrument. Peptides eluted from the liquid chromatography gradient were injected online to the mass spectrometer (lock mass enabled, mass range 400–1800 Da, resolution 60,000 at 400 Da, 10 MS/MS spectra per cycle, collision-induced dissociation at 35% normalised collision energy (CE), rejection of singly charged ions). The post-run raw data files were converted into .mgf files using the freeware MSConvert provided by Proteowizard (Chambers et al., 2012). Mgf files were analysed using PEAKS Studio 10.5 software (Ma et al., 2003), including the variable modifications ethylation (28.03 at D or E), carboxyethylation (72.02

at K or protein N-terminal groups), oxidation (M), acetylation (N-terminal) and the fixed modification carbamidomethyl (C). The data were refined using a false discovery rate of 1%.

## HeLa cell culture and transfection

HeLa DH (ECACC) cells were grown in Dulbecco's modified Eagle's medium (DMEM) supplemented with 10% fetal bovine serum, 50 µg/ml penicillin/streptomycin and 3 mM $CaCl_2$. For electrophysiology and intercellular dye transfer experiments, cells were seeded onto coverslips in six well plates at a density of $2 \times 10^4$ cells per well. After 24 h, the cells were transiently transfected with Cx26 constructs tagged at the C-terminus with a fluorescent marker (mCherry) according to the GeneJuice Transfection Reagent protocol (Merck Millipore).

## Cx26 and mutants

The sequence for human Cx26 (hCx26) and chicken Cx26 (GgCx26) were based on the sequences specified by accession numbers NP_003995 and NP_001257745.1, respectively. The mutations used in this study were introduced into the Cx26 gene by QuikChange site-directed mutagenesis and have been described previously (Meigh et al., 2013) (Cook et al., 2019). To generate chimeric hCx26-GgCx26 constructs, sequences were synthesised as DNA g-blocks (IDT) and double digested with KpnI and SmaI. Inserts were then gel purified prior to ligation into the corresponding sites of pCAG-mCherry. Following transformation into NEB5a cells, colonies were screened by PCR using a primer specific to the chicken–human interface, present only in the correct clones. Plasmid DNA was purified from positive PCR clones and the sequence confirmed by DNA sequencing (Source Bioscience).

## Solutions used

Control (35 mmHg $P_{CO_2}$) aCSF: 124 mM NaCl, 3 mM KCl, 2 mM $CaCl_2$, 26 mM $NaHCO_3$, 1.25 mM $NaH_2PO_4$, 1 mM $MgSO_4$, 10 mM D-glucose saturated with 95% $O_2$/5% $CO_2$, pH 7.4, $P_{CO_2}$ 35 mmHg.

20 mmHg aCSF: 140 mM NaCl, 3 mM KCl, 2 mM $CaCl_2$, 10 mM $NaHCO_3$, 1.25 mM $NaH_2PO_4$, 1 mM $MgSO_4$, 10 mM D-glucose, saturated with ~2% $CO_2$ (with the balance being $O_2$) to give a pH of 7.4 and a $P_{CO_2}$ of 20 mmHg, respectively.

55 mmHg aCSF: 100 mM NaCl, 3 mM KCl, 2 mM $CaCl_2$, 50 mM $NaHCO_3$, 1.25 mM $NaH_2PO_4$, 1 mM $MgSO_4$, 10 mM D-glucose, saturated with ~9% $CO_2$ (with the balance being $O_2$) to give a pH of 7.4 and a $P_{CO_2}$ of 55 mmHg, respectively.

70 mmHg aCSF: 73 mM NaCl, 3 mM KCl, 2 mM $CaCl_2$, 80 mM $NaHCO_3$, 1.25 mM $NaH_2PO_4$, 1 mM $MgSO_4$, 10 mM D-glucose, saturated with ~12% $CO_2$ (with the balance being $O_2$) to give a pH of 7.4 and a $P_{CO_2}$ of 70 mmHg, respectively.

## 3,3′-dioctadecyloxacarbocyanine, perchlorate (DiO) staining and expression analysis

HeLa DH (ECACC Cat# 96 112 022, RRID:CVCL_2483) were seeded onto glass coverslips at a density of $4 \times 10^5$ cells per well (6-well plate) and transfected 24 h later with the desired Cx26 construct using GeneJuice (Merck Cat#70 967-3). After 48 h, cells were washed three times with PBS and then fixed with 4% paraformaldehyde in PBS for 30 min before being washed three times with PBS. Cells were then incubated with serum-free DMEM containing 2.5 µM DiO Sigma-Aldrich Cat#D4292) for 15 min. After three washes in PBS, coverslips were mounted inverted on glass microscope slides in a Fluorshield with DAPI mounting medium (Sigma-Aldrich, Cat# F6057). Slides were sealed and imaged on a Zeiss 880 LSM confocal microscope using the 488 and 561 nm excitation wavelengths for DiO and mCherry, respectively.

Subsequent analysis was performed in Fiji, using the JaCoP plugin (Bolte & Cordelières, 2006). Regions of interest (ROIs) were drawn around mCherry positive cells and the image surrounding the ROI was removed. Manders' coefficient (Manders et al., 1993) was used as a quantitative assessment of colocalisation of mCherry with DiO, and hence the membrane localisation of the Cx26 construct. Thresholds were set to an appropriate limit that included DiO membrane staining but not any diffuse background.

## Dye loading

HeLa cells expressing WT or mutated Cx26 for 48 h were initially washed with a control solution (35 mmHg $P_{CO_2}$). They were then exposed to a test solution containing 20, 35, 55 or 70 mmHg $P_{CO_2}$ and 200 µM 5(6)-carboxyfluorescein (CBF) for 10 min. Subsequently, cells were returned to the control solution with 200 µM CBF for 5 min before being washed in the control solution without CBF for 30 min to remove the excess extracellular dye. A separate coverslip of HeLa cells was used for each condition tested. For each coverslip, mCherry staining was imaged to verify Cx26 expression. The experiments were replicated independently (independent transfections) at least five times for each Cx26 variant (to give $n$ = number of replicates for the dye-loading assay). Images were taken for each coverslip and each condition, such that at a later date, and by a person blinded to the experimental condition, the median pixel intensity for each expressing cell in the field of view could be measured

via ImageJ. Using ImageJ, the extent of dye loading was measured by drawing an ROI around individual cells and calculating the mean pixel intensity for the ROI. The mean pixel intensity of the background fluorescence was also measured in a representative background ROI, and this value was subtracted from the measures obtained from the cells. For each coverslip a minimum of 40 cells were measured. The medians of these measurements were then used as the quantitated variable for subsequent statistical analysis.

### Imaging assay of gap junction transfer

2-Deoxy-2-[(7-nitro-2,1,3-benzoxadiazol-4-yl)amino]-D-glucose (NBDG) was included at 200 μM in the patch recording fluid, which contained: K-gluconate 130 mM; KCl 10 mM; EGTA 5 mM; CaCl$_2$ 2 mM, HEPES 10 mM; pH was adjusted to 7.3 with KOH to give a resulting final osmolarity of 295 mOsm. Cells were imaged on a Cleverscope (MCI Neuroscience) with a Photometrics Prime camera under the control of Micromanager 1.4 software. LED illumination (Cairn Research) and an image splitter (Optosplit, Cairn Research) allowed simultaneous imaging of the mCherry-tagged Cx26 subunits and the diffusion of the NBDG into and between cells. Coupled cells for intercellular dye transfer experiments were initially selected based on tagged Cx26 protein expression and the presence of a gap junctional plaque, easily visible as a band of mCherry fluorescence (e.g. Figs. 8 and 9). After establishing the whole-cell mode of recording, images were collected every 10 s. The assay was performed as described by Nijjar et al. (2021). The start of recording ('time zero') was taken to be the first image following the establishment of a stable whole-cell recording. Dye permeation between cells was measured either at 35 mmHg aCSF for 10 min, or 55 mmHg aCSF for 2 min (WT gap junction shut) followed by a switch to 35 mmHg aCSF to open the gap junction. In cases where permeation occurred within 2 min in 55 mmHg aCSF, the switch to 35 mmHg was deemed unnecessary as the gap junction was already open (e.g. Figs. 9–11). For each gap junction recording (considered as an independent statistical replicate), analysis of the cell images was performed in ImageJ using an ROI drawn on each cell to measure the median pixel intensity of the donor and acceptor cells. The time for the median fluorescence of the acceptor cell to reach 10% of the donor cell was then determined and used as the value for statistical comparisons.

### Measurement of ATP release and analysis

pDisplay-GRAB_ATP1.0-IRES-mCherry-CAAX was a gift from Yulong Li (Addgene plasmid #167 582; http://n2t.net/addgene:167583; RRID: Addgene_167 582).

ATP release experiments were performed as per a previous study (Butler & Dale, 2023). In brief, cells were imaged by epifluorescence (Scientifica Slice Scope, Cairn Research OptoLED illumination, 60× water Olympus immersion objective, NA 1.0, Hamamatsu ImagEM EM-SSC camera, Metafluor software). The cpGFP of GRAB$_{ATP}$ was excited using 470 nm LED, with fluorescent emission being recorded between 507 and 543 nm. The mCherry-tagged Cx26 constructs were excited with the 535 nm LED, with emission being recorded between 570 and 640 nm.

Analysis of GRAB$_{ATP}$ signals was performed in ImageJ. For cells that expressed both the Cx26 variant and GRAB$_{ATP}$, an ROI was drawn around the region of GRAB$_{ATP}$ expression in each cell and the median pixel intensity within the ROIs measured for each image. The fluorescence pixel intensities (F) were normalised to a baseline period (F$_0$), and the difference in F/F$_0$ (ΔF/F$_0$) evoked by the CO$_2$ stimulus measured for each cell. As the concentration–response curve for GRAB$_{ATP}$ was approximately linear over the range 0–3 μM, changes of fluorescence evoked by changes in $P_{CO_2}$ were converted into ATP concentration by normalising them to the ΔF/F$_0$ produced by a 3 μM calibration dose of ATP in each experiment. Statistical comparisons were performed considering each cell as an independent measurement. At least three independent transfections were performed for each variant of Cx26.

### Patch clamp recording

Coverslips containing non-confluent HeLa cells were placed into a perfusion chamber at room temperature and superfused with standard aCSF. An Axopatch 200B amplifier was used to make whole-cell recordings from single HeLa cells. The intracellular fluid in the patch pipettes contained: K-gluconate 130 mM, KCl 10 mM, EGTA 10 mM, CaCl$_2$ 2 mM, HEPES 10 mM, sterile filtered, pH adjusted to 7.3 with KOH. An agarose salt bridge was used to avoid solution changes altering the potential of the Ag/AgCl reference electrode. All whole-cell recordings were performed at a holding potential of -50 mV. Whole-cell conductance was measured by repeated steps to -40 mV. To allow for any drift in whole-cell conductance unrelated to the CO$_2$ stimulus, the maximal conductance during the CO$_2$ test stimulus was compared to the average of the conductance before the stimulus and after the stimulus had been fully washed off. Each whole-cell recording was considered to be an independent statistical replicate.

### Cloning and purification of chicken Cx26

DNA encoding for chicken connexin 26 (GgCx26) lacking a stop codon was prepared by PCR

from a GgCx26-mCherry expression construct described previously (de Wolf et al., 2017), using primers which introduced a thrombin-cleavable C-terminal His-6 purification tag (forward primer: 5'-CACGACGAATTCCACCATGGATTGGGG 3', reverse primer 5'-GCTATCACTAGTCTTTAAAACTGT TG-3'). The amplified GgCx26 sequence was double digested with EcoRI and SpeI, prior to ligation into the corresponding sites of the pFastbac1 vector (NEB). The construct was sequence verified before being transformed into DH10Bac cells (Invitrogen) for bacmid production, and subsequent baculovirus production in Sf9 cells, according to the manufacturer's instructions. Sf9 cells were harvested 72 h post-infection for protein purification. The purification was performed as for hCx26 described previously (Brotherton et al., 2022), with the following changes: solubilisation was for 3.5 h, and dialysis was carried out against (10 mM sodium phosphate, 500 mM NaCl, 5% glycerol, 1 mM DTT, 0.03% DDM (Glycon Biochemicals GMBH), pH 7.4) to remove histidine, before flash-freezing the protein. On thawing

the protein for final preparation, the protein was digested with thrombin (1:1 w/w) (Sigma) and concentrated prior to size exclusion chromatography using a Superose 6 5/150 column (GE Healthcare Lifescience) to remove thrombin and exchange the buffer to 90 mmHg aCSF buffer (70 mM NaCl, 5% glycerol, 1 mM DTT, 0.03% DDM, 80 mM $NaHCO_3$, 1.25 mM $NaH_2PO_4$, 3 mM KCl, 1 mM $MgSO_4$, 4 mM $MgCl_2$.)

## Cryo-EM sample preparation and data collection

The GgCx26 peak was concentrated to 3.4 mg/ml before being gassed with the correct amount of 15% $CO_2$ to give a final pH of ~7.4 as previously described (Brotherton et al., 2022). Quantifoil 300 mesh gold grids 0.6/1 carbon film (Quantifoil Micro Tools GMBH) were glow-discharged at 3 mA, for 60 s prior to use. Vitrification of the protein in liquid ethane/propane at -180°C was carried out with a Leica GP2 automated plunge freezer with 3 ml protein per grid at 10°C, 71% humidity (set: 4°C, 95% humidity), 7 s blotting in a 15% $CO_2/N_2$ atmosphere. Grids were

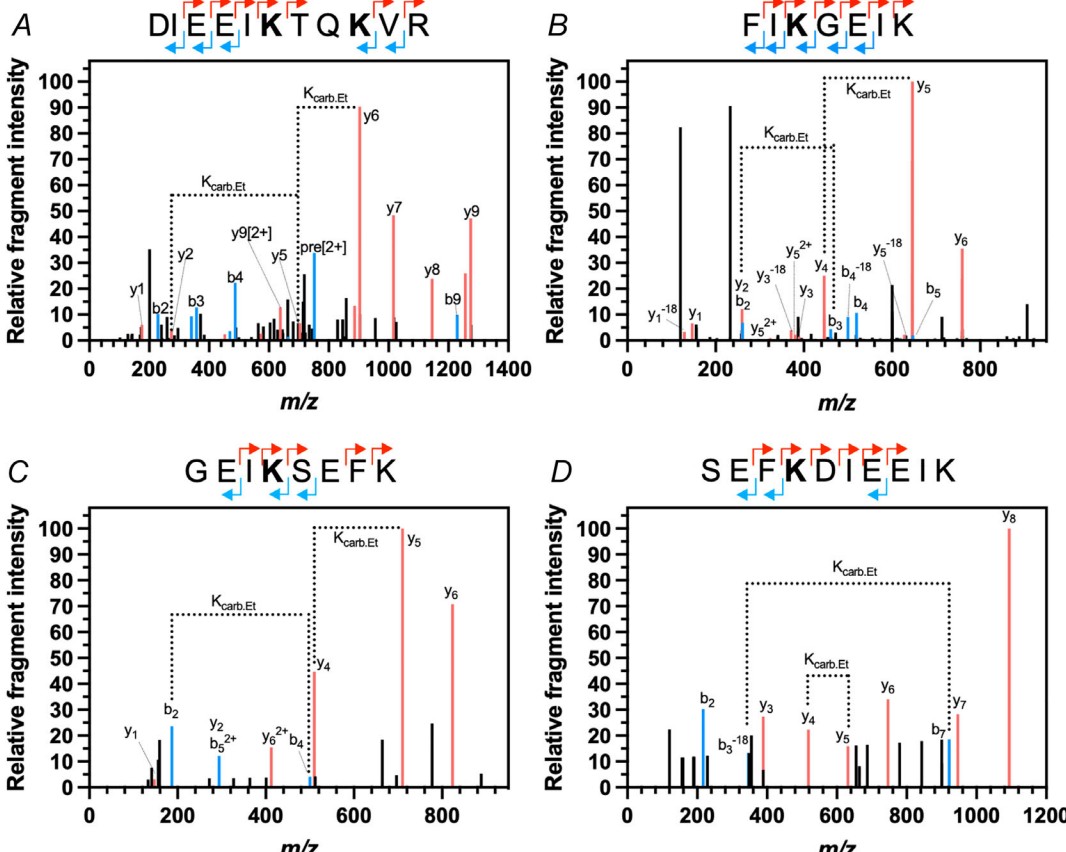

**Figure 1. CO$_2$ binds Cx26**

*A–D*, demonstration of exchangeable CO$_2$-binding sites on Cx26 by MS/MS. Plots of relative fragment intensity *versus* mass/charge ratio (m/z) for fragmentation data from MS/MS identifying ethyl-trapped carbamate on K122 and K125 (*A*), K108 (*B*), K112 (*C*) and K116 (*D*), of recombinant human Cx26 protein. Peptide sequences indicate predominant +1y (red) +1b (blue) ions identified by MS/MS shown in the plot. The modified residue is indicated in bold. K$_{carb.Et}$ indicates the molecular weight difference between ions diagnostic of the modified Lys. [Colour figure can be viewed at wileyonlinelibrary.com]

screened using a Jeol 2100plus microscope, and data were collected on an FEI Titan Krios G3 equipped with a K3 detector and BioQuantum energy filter using a 20 eV slit width. A dose rate of ∼10 e/pix/s on the detector was chosen with a final dose of 46 e/$\text{Å}^2$ on the specimen. Data collections were prepared and run automatically using EPU2.1 and aberration-free image shift.

### Cryo-EM data processing

Data were processed using Relion3.1-beta (Zivanov et al., 2018). Micrographs were motion corrected using the version of MotionCor2 (Zheng et al., 2017) implemented in Relion, and CTFs were estimated using CTFFIND4 (Rohou & Grigorieff, 2015). Particles were picked using the Laplacian of Gaussian picker, and after three rounds of 2D classifications with particles down-sampled to 4 Å/pixel, good particles were selected. 3D classification was carried out in C1 with an initial model generated from a previous Cx26 cryo-EM structure (unpublished data) with a low-pass filter of 60 Å. Multiple rounds of 3D classification in C1 resulted in two very similar classes, which were selected for continued processing. The unbinned particles from these two classes were subjected to refinement, CTF refinement and polishing in Relion to improve the resolution of the Coulomb shells until no further improvement was gained. The resolution was estimated based on the gold standard Fourier shell

correlation (FSC) criterion (Rosenthal & Henderson, 2003; Scheres, 2012) with a soft solvent mask. Local resolution estimation was carried out in Relion. Masks for processing were prepared in Chimera (Goddard et al., 2007; Pettersen et al., 2004).

### Model building

Initial model construction by sequence modification from human 90 mmHg Cx26 (Brotherton et al., 2022) and building was carried out in Coot (Emsley & Cowtan, 2004). Density for the N-terminal helix was not well defined, and residues were not included before Val 9. Residues between 106 and 124 in the cytoplasmic loop and the C-terminus after residue Leu215 were also omitted due to lack of evident density. Real space refinement in Phenix (Liebschner et al., 2019) was carried out with non-crystallographic symmetry constraints. The region between 36 and 43 was modelled in two conformations. Water molecules were added into spherical density and where lipids or detergents were clearly visible in the density, these have been modelled as hydrocarbon chains.

### Structural analysis

Structural images shown in this paper were generated in Chimera and PyMol. Superpositions were carried out in Chimera such that only matching $C_a$ pairs within 2 Å after superposition were included in the matrix calculation.

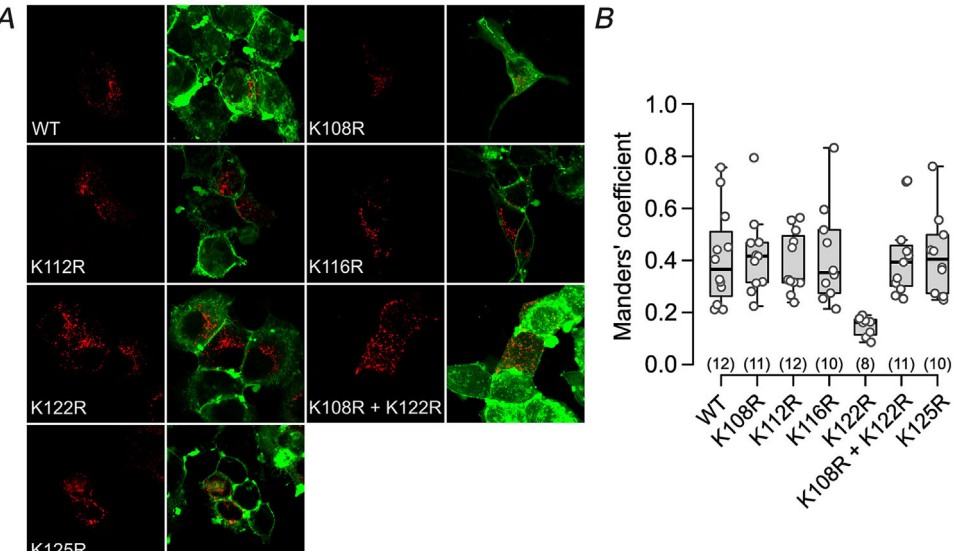

**Figure 2. Confocal imaging of expression patterns of wild-type (WT) Cx26 and carbamylation mutants**
*A*, single optical sections showing expression patterns in HeLa cells of wild-type Cx26 (WT) tagged with mCherry and Cx26 mutations of the lysine residues (Lys to Arg) identified as being subject to carbamylation. A double mutation for K108 and K122 was also performed. The cell membrane was counterstained with the lipophilic dye DiO. Scale bar 20 µm. *B*, quantification of mCherry and DiO colocalisation. All mutants apart from K122R exhibited the same amount of colocalisation as measured by the Manders' coefficient as the WT. Numbers in brackets are the number of cells analysed. Kruskal–Wallis ANOVA ($dF = 6$, $P = 0.0016$) and Mann–Whitney test WT *vs.* K122R ($P = 0.00025$). [Colour figure can be viewed at wileyonlinelibrary.com]

## Statistical presentation and analysis

For the concentration–response curves, the median and first and third quartiles are plotted. The fitted curve is the Hill equation appropriately scaled, with the Hill coefficient and $EC_{50}$ chosen for the most plausible fit. All other quantitative data are presented as box and whisker plots where the box represents the interquartile range, the bar represents the median, and the whiskers represent 1.5 times the interquartile range. Individual data points are superimposed. Statistical analysis was via the Kruskal–Wallis one-way ANOVA (Figs. 2–6) followed by pairwise Mann–Whitney U tests with correction for multiple comparisons via the false discovery method (Curran-Everett, 2000) with a maximum rate of false discovery set at 0.05. Comparisons in Figs. 8–11 were via the Mann–Whitney U test. All pairwise comparisons were two-sided.

## Results

### Multiple lysine residues are carbamylated within the cytoplasmic loop

Protein carbamylation can be directly detected by covalent trapping with the alkylation reagent TEO. To investigate the presence of lysine carbamylation on Cx26, purified, DDM-solubilised hCx26 in buffer corresponding to a $P_{CO_2}$ of 90 mmHg and at a pH of 7.4, below the unmodified pKa of lysine, was treated with TEO to trap $CO_2$. This sample was digested with trypsin. Trypsin cleaves at the C-terminal side of positively charged Lys and Arg residues. However, modification of the lysine by carbamylation will prevent cleavage by the protease resulting in a missed cleavage. Using ESI-MS/MS, five residues were observed to have avoided cleavage and have the additional mass associated with the alkylated carbamate group. This included Lys125 (Fig. 1A), which had been predicted to be the $CO_2$ binding site, and four other lysine residues on the same cytoplasmic loop (Lys108, Lys112 and Lys116 and Lys122; Fig. 1A–D).

At a $P_{CO_2}$ of 20 mmHg and pH 7.4, no carbamylation of K108, K122 or K125 was observed, but K112 and K116 were clearly carbamylated at this low level of $CO_2$. hCx26, therefore, binds $CO_2$ by lysine carbamylation, and this carbamylation event is labile and sensitive to environmental $P_{CO_2}$. Further, the hCx26 lysine carbamylation profile is more complex than previous mutagenesis studies suggest. The carbamylation of these five lysine residues within the cytoplasmic loop suggests that they share a common environment that favours this post-translational modification by $CO_2$.

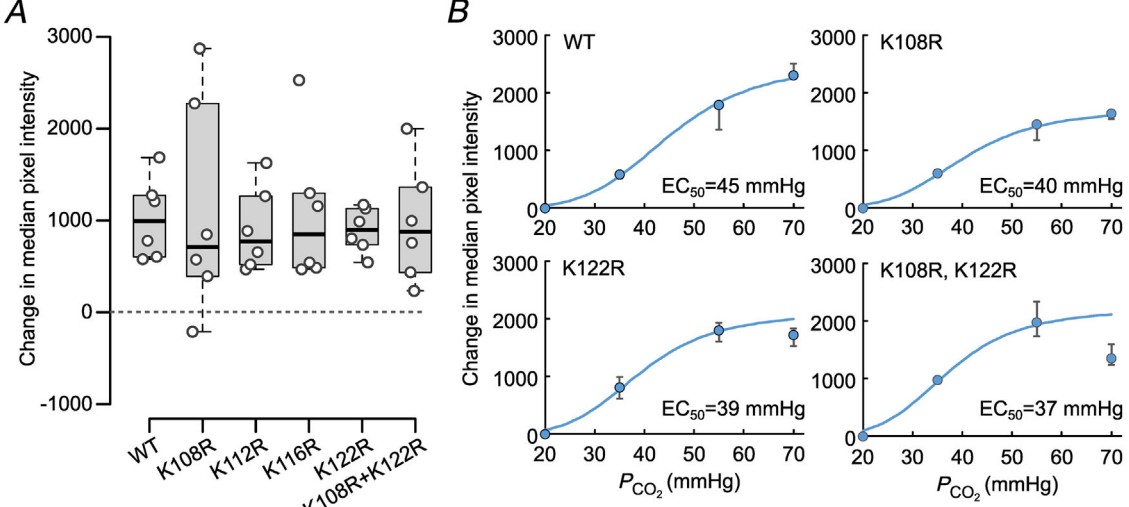

**Figure 3. Carbamylation of additional lysine residues in the cytoplasmic loop is not required for $CO_2$-dependent hemichannel opening**

*A*, for all mutations, the amount of dye loading in response to a stimulus of 55 mmHg $P_{CO_2}$ (from 35 mmHg, both solutions pH 7.4) was similar to wild-type Cx26. Each point on the graph shows the difference in median fluorescence between cells exposed to carboxyfluorescein at 35 mmHg $P_{CO_2}$ and 55 mmHg $P_{CO_2}$ from six independent transfections with at least 200 cells analysed per condition per transfection to generate the median fluorescence value for each condition. There were no significant differences between groups (Kruskal–Wallis test, $P = 0.9885$, d$F = 5$). *B*, $CO_2$ concentration-dependent changes in dye loading from 20 to 70 mmHg (the pixel intensity at 20 mmHg was set to zero, and the changes to higher $P_{CO_2}$ expressed relative to this) for Cx26$^{WT}$, Cx26$^{K108R}$, Cx26$^{K122R}$ and Cx26$^{K108R,K122R}$. The blue line is a Hill equation with the indicated $EC_{50}$ and a Hill coefficient of 5 in all cases. Note that K122R causes a drop in dye loading at 70 mmHg, which is even more marked for the double mutation. Five independent repeats for each level of $P_{CO_2}$. Data expressed as medians with the lower and upper quartiles indicated. [Colour figure can be viewed at wileyonlinelibrary.com]

## Additional carbamylation sites are not required for CO$_2$-dependent hemichannel opening

Our prior work had identified and tested the essential role of Lys125 for hemichannel opening. As additional lysine residues can be carbamylated, we tested whether these might also be necessary for hemichannel opening. We, therefore, mutated these lysine residues individually to arginine, which cannot be carbamylated.

To assess the possibility that these K to R mutations might alter the pattern of expression of Cx26, we stained the plasma membrane of mutant-expressing HeLa cells after fixation with the lipophilic dye DiO. The colocalisation mCherry-tagged mutants with the DiO-stained membrane was quantified and used to derive Manders' coefficients of colocalisation (Fig. 2).

The mutations K108R, K112R, K116R, K125R and K108R+K122R had Manders' coefficients (median and 95% confidence interval) of, respectively: 0.42, 0.28–0.54; 0.33, 0.31–0.52; 0.35, 0.25–0.60; 0.4, 0.26–0.56; and 0.39, 0.27–0.7. These Manders' coefficients were unchanged from those for WT Cx26 (0.37, 0.23–0.57). By contrast, the mutation K122R depressed localisation of Cx26 at the membrane (0.16, 0.09–0.19; $P = 0.00025$ compared with WT, Mann–Whitney's test).

Using a dye-loading assay, we found that none of these lysine residues were essential for hemichannel opening (Fig. 3*A*, Kruskal–Wallis test, $P = 0.9885$, d$F = 5$). As only some of the lysine residues showed concentration-dependent carbamylation when analysed by ESI-MS/MS (K108, K122 and K125), we examined whether these might influence the sensitivity of hCx26

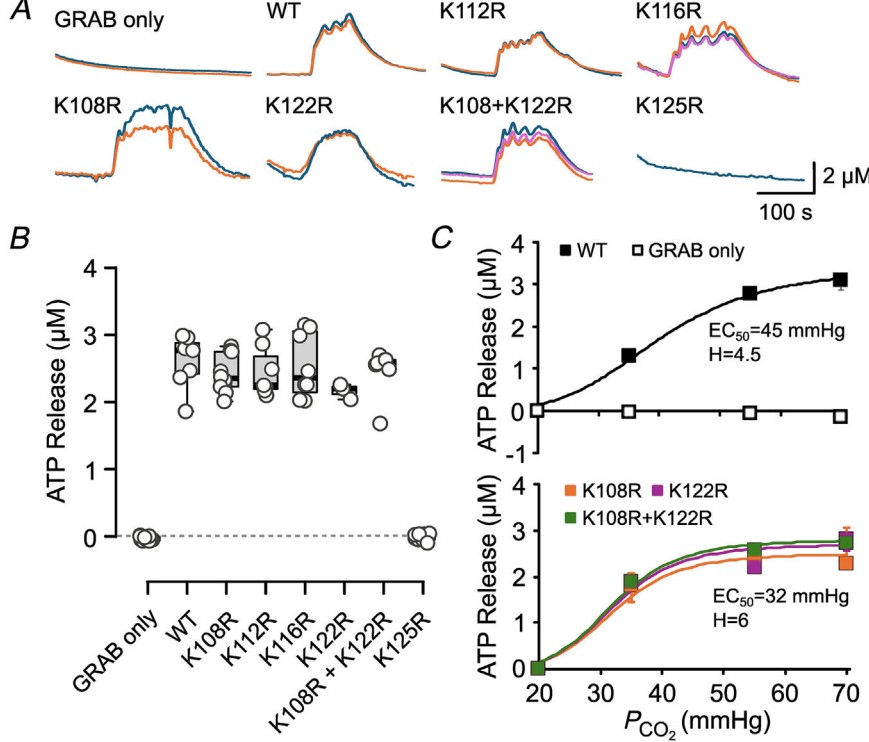

**Figure 4. Carbamylation of K125R, but not other lysine residues, is required for CO$_2$-dependent ATP release via Cx26 hemichannels**

*A*, sample traces of changes in GRAB$_{ATP}$ fluorescence in response to a change from 20 to 70 mmHg $P_{CO_2}$ for each mutant except K112R and K116R where the change was from 20 to 55 mmHg. *B*, summary data showing that CO$_2$-dependent ATP release (evoked by a change from 20 to 55 mmHg) is the same in the wild-type hemichannel and all mutants apart from K125R (combined data from three independent transfections). HeLa cells transfected only with the GRAB$_{ATP}$ sensor do not exhibit CO$_2$-dependent ATP release. A Kruskal–Wallis ANOVA shows that these samples are not drawn from the same population ($P = 2.07 \times 10^{-7}$, d$F = 7$). Pairwise comparison reveals that GRAB only and Cx26$^{K125R}$ differ from Cx26$^{WT}$ ($P = 0.0004$ and $P = 0.001$, respectively). *C*, concentration–response curves for ATP release from HeLa cells expressing Cx26$^{WT}$, Cx26$^{K108R}$, Cx26$^{K122R}$, Cx26$^{K108R,K122R}$ and GRAB only. While carbamylation of Lys108 and Lys122 is not required for hemichannel opening their mutation to Arg shifts the CO$_2$ concentration–response curve to lower levels of $P_{CO_2}$. The continuous curves are drawn according to the Hill equation with the parameters given on the graphs. Data expressed as the median with the lower and upper quartiles indicated. GRAB only $n = 12$ cells; Cx26$^{WT}$ $n = 7$ cells; Cx26$^{K108R}$ $n = 10$ cells; Cx26$^{K112R}$ $n = 8$ cells; Cx26$^{K116R}$ $n = 8$ cells; Cx26$^{K122R}$ $n = 3$ cells; Cx26$^{K108R,K122R}$ $n = 7$ cells; and Cx26$^{K125R}$ $n = 9$ cells. [Colour figure can be viewed at wileyonlinelibrary.com]

hemichannels to different levels of $P_{CO_2}$. Individually, K108R and K122R appeared to slightly shift the concentration–response curve to lower levels of $P_{CO_2}$, and the double mutation K108R-K122R accentuated this shift (Fig. 3*B*). This suggests that these mutations may subtly increase the sensitivity of the hemichannel to $CO_2$. The mutation K122R also decreased the dye loading at the highest level of $P_{CO_2}$ (70 mmHg), which was even more marked when combined with K108R. Nevertheless, unlike K125, which is essential for hemichannel opening (Meigh et al., 2013; van de Wiel et al., 2020), carbamylation of these other lysine residues is not required for the $CO_2$-induced opening of hCx26 hemichannels.

As further confirmation of this key result, we used a newly developed real-time assay of ATP release to measure the $CO_2$ dependence of Cx26$^{WT}$ and the carbamylation mutants (Butler & Dale, 2023). Parental HeLa cells that did not express Cx26$^{WT}$ did not exhibit $CO_2$-dependent ATP release (Fig. 4*A,B*). However, CX26$^{WT}$-expressing cells, and cells that expressed all of the carbamylation mutants apart from Cx26$^{K125R}$ released ATP in response to an elevation of $P_{CO_2}$ from 20 to 55 mmHg. We analysed the concentration dependence of ATP release for mutations of the residues that undergo concentration-dependent carbamylation when analysed by ESI-MS/MS. We found that the mutations K108R, K122R and K108R+K122R shifted the concentration dependency of ATP release to lower levels of $P_{CO_2}$ (Fig. 4*C*).

We next used patch-clamp recording to investigate the gating of Cx26 by $CO_2$. We confirmed that parental HeLa cells do not exhibit a $CO_2$-dependent conductance change on going from 35 to 55 mmHg $P_{CO_2}$ (Fig. 5), and that

the change in whole-cell conductance evoked by $CO_2$ in Cx26-expressing HeLa cells can be blocked by 200 μm LaCl$_3$ (Fig. 5). We investigated the $CO_2$ sensitivity of the Cx26 lysine mutants. As with the dye-loading and ATP-release assays, neither K108 nor K122 were required for hemichannel opening (Fig. 6). Furthermore, the double mutant K108R, K122R retained $CO_2$ sensitivity. By contrast, the mutation K125R abolished $CO_2$ sensitivity, as documented previously (Fig. 6; $P = 2.2 \times 10^{-6}$ for 35 mmHg *vs*. WT and $P = 0.008$ for 55 mmHg *vs*. WT).

## CO$_2$-dependent gap junction channel closure

$CO_2$ has two effects on Cx26. In addition to increasing hemichannel gating, elevated $P_{CO_2}$ closes gap junction channels (Nijjar et al., 2021). Consistent with this, a comparison of the structures of human Cx26 gap junction channels at low *vs*. high $P_{CO_2}$, demonstrates a more constricted pore at high $P_{CO_2}$ than seen at low $P_{CO_2}$ at the same pH (Brotherton et al., 2022). Structural studies of connexins have been hampered by the absence of density for the cytoplasmic loop on which the lysines discussed above reside. In an effort to obtain a structure of Cx26 where the loop is visible, we solved the structure of GgCx26, which has a cytoplasmic loop that is shorter by four residues, using cryo-EM (Tables 1 and 2). GgCx26 has the carbamylation motif, and GgCx26 hemichannels are opened by $CO_2$ (de Wolf et al., 2017). Using conditions similar to those used to obtain the structure of hCx26 at high $P_{CO_2}$, reconstructions were obtained at a resolution of 2.1 Å (Fig. 7). Overall, the structure is very similar to the hCx26 gap junction channel structures including

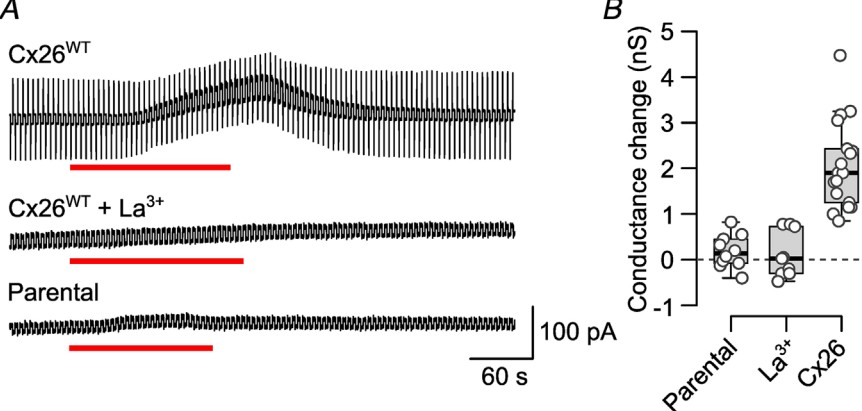

**Figure 5. CO$_2$-dependent conductance change in HeLa cells depends on the presence of Cx26, and can be blocked by La$^{3+}$**
*A*, sample records showing whole-cell conductance change evoked in parental (untransfected) HeLa cells, and HeLa cells transiently expressing Cx26$^{WT}$ with or without the presence of 200 μM LaCl$_3$. Red bar indicates application of 55 mmHg $P_{CO_2}$ from a baseline of 35 mmHg. *B*, summary data showing that Cx26-expressing HeLa cells show a greater conductance change in response to 55 mmHg $P_{CO_2}$ than parental cells (Mann–Whitney test, $P = 1.768 \times 10^{-5}$) and from Cx26-expressing cells in the presence of La$^{3+}$ (Mann–Whitney test, $P = 3.428 \times 10^{-5}$). A Kruskal–Wallis ANOVA shows that these samples were not drawn from the same population ($P = 1.3 \times 10^{-6}$, d$F = 2$). [Colour figure can be viewed at wileyonlinelibrary.com]

having lipid-like density within the pore (Fig. 7). The replacement of an alanine with a glutamine at residue 49 towards the extracellular side of the pore causes a slightly narrower pore (14 Å) but would not be sufficiently narrow to restrict access to small molecules (Fig. 7). We previously commented on the difference between the crystal structures and cryo-EM structures of hCx26 at residues 36–42 at the extracellular side of TM1 that results in the rotation of Ala40. In the GgCx26 structure there is evidence of both conformations giving support to this being a flexible region of the protein (Fig. 7). As for other connexin structures, no density was apparent for the cytoplasmic loop between residues Arg 105 and Arg 125 (equivalent to Lys105 and Glu129 of hCx26, respectively) (Flores et al., 2020; Lee et al., 2020; Qi et al., 2022; Qi et al., 2023; Yue et al., 2021). The major surprise, however, was that when compared with the hCx26 structures, the map, with an ill-defined N-terminal helix, more closely resembled the map derived from low $P_{CO_2}$, which was attributed to the open conformation, rather than high $P_{CO_2}$, which was more constricted (Fig. 7) (Brotherton et al., 2022). In the GgCx26 structure, the linking region between the N-terminus and TM1 is better defined than seen for the low $P_{CO_2}$ hCx26 structure (Fig. 7C). It is

**Table 1. Cryogenic electron microscopy data collection and processing statistics**

| | |
|---|---|
| Voltage (kV) | 300 |
| Magnification (×1000) X | 105 |
| Camera | K3 |
| Camera mode | Super-resolution |
| Energy filter (eV) | 20 |
| Defocus range (mm) | −2 to −0.8 |
| Pixel size (Å/pix) | 0.85 |
| Dose on detector (e⁻/pix/s) | 10 |
| Dose on sample (e⁻/pix/s) | 11 |
| Exposure time | 3 |
| No. of images | 8713 |
| Frames per image | 45 |
| Initial particle number | 2,041,681 |
| Final particle number | 162,951 |
| **Resolution**[a] | |
| masked D6 | 2.1 |
| masked C6 | 2.1 |

[a] From Relion_postprocess (Zivanov et al., 2018).

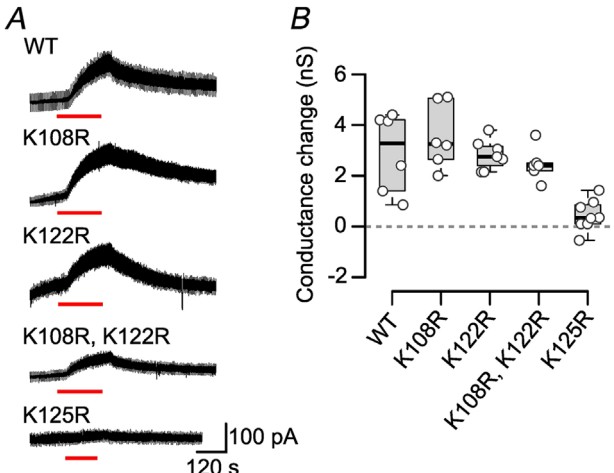

**Figure 6. Carbamylation of Lys125 is essential for the CO₂-dependent change in whole-cell conductance in Cx26-expressing cells**

*A*, representative whole-cell current records showing the effect of a change in $P_{CO_2}$ (indicated by red bars) from 35 to 55 mmHg. Recordings from HeLa cells expressing the wild-type or mutant channels. The cells were recorded at a holding potential of -50 mV with regular steps to -40 mV to measure whole-cell conductance. Only the mutation K125R blocked the CO₂ sensitivity of the hemichannels. *B*, summary data for the whole-cell conductance change observed for an increase in $P_{CO_2}$ from 35 to 55 mmHg (Kruskal–Wallis test, P = 0.001, dF = 4). The mutation K125R effectively abolishes the change in conductance induced by CO₂ (Mann–Whitney test: 35–55 mmHg P = 0.008 compared with wild-type). [Colour figure can be viewed at wileyonlinelibrary.com]

located more towards the cytoplasmic side than seen for the hCx26 structures so would pull the N-terminal helix back from the pore as suggested previously.

The apparently open structure of the GgCx26 gap junction channel at high $P_{CO_2}$ prompted us to investigate whether GgCx26 gap junction channels are indeed closed with increases in $P_{CO_2}$. We, therefore, used a dye transfer assay and measured the length of time it took for the dye to transfer between coupled cells at different levels of $P_{CO_2}$ (Nijjar et al., 2021). In human Cx26, at a $P_{CO_2}$ of 55 mmHg, the gap junction is closed, and no dye permeates until after the $P_{CO_2}$ is reduced to 35 mmHg (Nijjar et al., 2021). For GgCx26 gap junction channels, dye permeated between coupled cells rapidly at both 35 mmHg and 55 mmHg $P_{CO_2}$. Thus, unlike human Cx26, the GgCx26 gap junction channel is insensitive to modest changes in $P_{CO_2}$ (Fig. 8).

Lys 108 is positioned on the second transmembrane helix, in the next turn above the critical Arg 104 (Brotherton et al., 2024; Maeda et al., 2009). Sequence comparisons of human Cx26 and GgCx26 revealed that the equivalent residue to Lys108 of human Cx26 in GgCx26 is Ser108. As the MS/MS screen revealed that Lys108 is differentially carbamylated, we hypothesised that due to the similar position to Arg104, Lys108 might be essential for gap junction closure by CO₂. We, therefore, investigated the effect of the mutation K108R on the CO₂ sensitivity of human Cx26 gap junction channels. This mutation removed the CO₂ sensitivity from human Cx26 gap junction channels (Fig. 9). Thus, in contrast to hemichannel opening, where only carbamylation of K125 is required, both K108 and K125 are required to be carbamylated for gap junction closure.

We next tested whether the mutation of S108K in GgCx26 might restore gap junction channel sensitivity, but this did not occur (Fig. 10). Substitution of a large part of the cytoplasmic loop (the missing residues from the crystal structure) from hCx26 in GgCx26 did not restore the $CO_2$ sensitivity of the chimeric gap junction channel. We also inserted the entire cytoplasmic loop from hCx26 (including the carbamylation motif) into GgCx26. This also failed to restore $CO_2$ sensitivity of the chimeric gap junction channel. This would suggest that the interacting partner for K108 does not reside within the human cytoplasmic loop. Nevertheless, for both cytoplasmic loop substitutions, the chimeric hemichannel still responded to $CO_2$ (Fig. 10), showing that insertion of the human cytoplasmic loop in place of that of the chicken did not disrupt $CO_2$ sensitivity of the hemichannel.

## R216: a potential interacting partner for K108

Given that none of the reported connexin structures contained the residues of the cytoplasmic loop, we inspected the Alphafold 2 predicted structure for Cx26. Within the more confidently predicted residues, R216 on TM4 was identified as a potential interaction partner. It should be noted that in the most recent experimentally derived structure of hCx26 gap junction channels at high $P_{CO_2}$, these residues do not appear to interact (Brotherton et al., 2024). Nevertheless, we made the mutation R216Q to remove the potential for formation of this salt bridge. This mutation completely blocked $CO_2$-dependent gap junction channel closure (Fig. 11*A*,*B*) but did not prevent $CO_2$-dependent opening of the hemichannel (Fig. 11*C*,*D*).

We further tested the potential for interaction between K108 and R216 via the double mutation K108R-R216E.

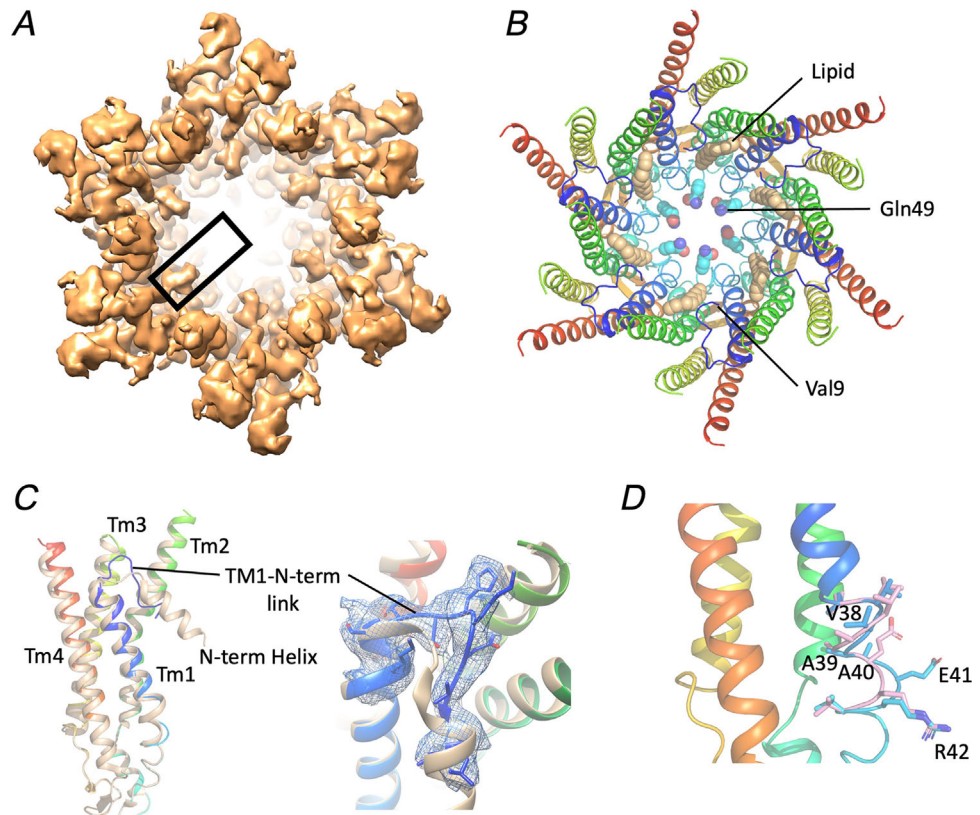

**Figure 7. Cryo-EM structure for GgCx26**

*A*, Coulomb shell viewed from the cytoplasmic side illustrating that the N-terminal helix is ill-defined. The expected position for the N-terminus is marked by a black rectangle in one subunit. *B*, ribbon illustration of the structure in a similar view to *A*. The residues are coloured according to residue number with blue at the N-terminus and red at the C-terminus. Gln49, which forms the narrowest part of the pore is shown as spheres. The putative lipid on the pore wall is shown with coral spheres. *C*, comparison with hCx26. Left, one subunit of GgCx26 (coloured as in *B*) superposed on a subunit of the hCx26 at a $P_{CO_2}$ of 90 mmHg (beige). Overall, the structures are similar with a root mean square deviation of 0.5 Å for 171 Ca pairs out of 184. The major difference is at the N-terminus. The linker between the N-terminus and TM1 is raised up towards the cytoplasmic side. Right, close up of the linker with the associated density. *D*, two conformations have been modelled at the TM1/extracellular loop interface. In one conformation this is more like the crystal structures, in the other this resembles the cryo-EM structures. [Colour figure can be viewed at wileyonlinelibrary.com]

Here, the introduction of a negative charge at residue 216 could force interaction with the positively charged R108 residue, to form a salt bridge in reverse. Prior studies have shown that reverse salt bridge formation can be effective: the mutation R104E gives constitutively open hemichannels (Meigh et al., 2013). We found that K108R-R216E gap junction channels were constitutively closed even at the control level of $P_{CO_2}$ (35 mmHg) and were insensitive to further increases of $CO_2$ (Fig. 12).

However, the hemichannels remained sensitive to $CO_2$ (Fig. 12).

## Discussion

A key result from this paper is that the cytoplasmic loop of Cx26 resides in a chemical environment that promotes spontaneous carbamylation of the lysine residues 108,

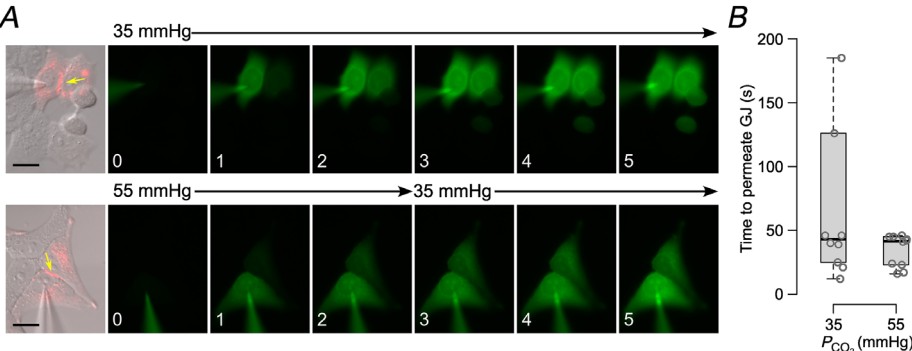

**Figure 8. GgCx26 gap junction channels are not sensitive to $CO_2$**
*A*, dye transfer assay between HeLa cells coupled by GgCx26. GgCx26 was tagged with mCherry, and the gap junction is visible in the DIC brightfield images as a red stripe, indicated by a yellow arrow. The fluorescence images show the transfer of NBDG from the cell loaded with the dye via a patch pipette to the acceptor cell. In the top row of images, the $P_{CO_2}$ of the solution was maintained at 35 mmHg. In the bottom row, $P_{CO_2}$ was at 55 mmHg for the first 2 min of the recording before being changed to 35 mmHg. Note that the dye has already permeated the acceptor cell within 2 min at 55 mmHg. Scale bars 20 μm. The numbers in the bottom left corners of the images represent the time of the image in minutes following the establishment of the whole-cell recording. *B*, the box plots show the time (from establishment of whole-cell recording) for the fluorescence of the acceptor cell to reach 10% of that of the donor cell for the two conditions. Mann–Whitney test, *P* = 0.344, 35 *vs.* 55 mmHg. [Colour figure can be viewed at wileyonlinelibrary.com]

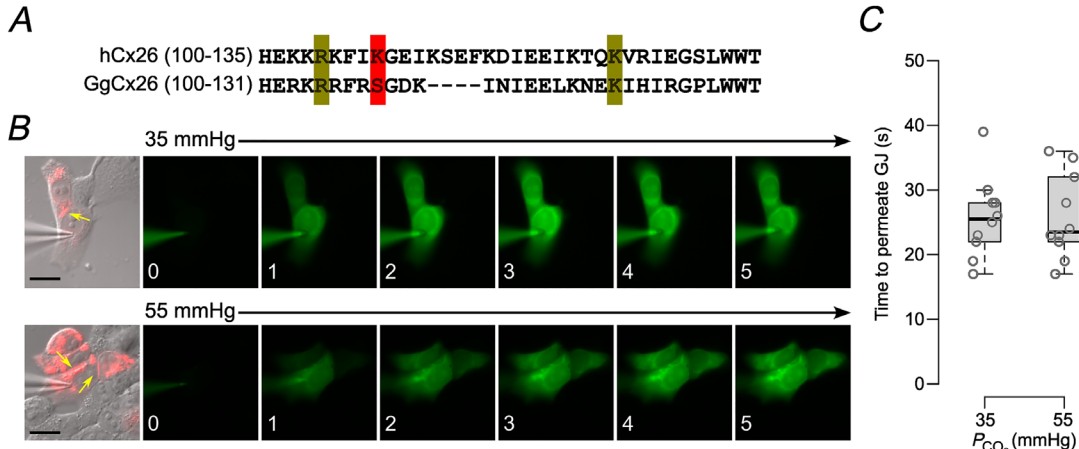

**Figure 9. K108R removes $CO_2$ sensitivity from hCx26 gap junction channels**
*A*, sequence comparison of the cytoplasmic loops of GgCx26 and hCx26. K125 and R104 and their equivalents are highlighted in olive, and K108 and S108 are highlighted in red. *B*, hCx26[K108R] tagged with mCherry was expressed in HeLa cells. Gap junctions are evident as a red stripe (yellow arrow). The permeation of dye from donor to acceptor cell was equally fast at 35 and 55 mmHg $P_{CO_2}$. The numbers in the bottom left corners of the images represent the time of the image in minutes following the establishment of the whole-cell recording. Scale bar 20 μm. *C*, summary data showing the time for the fluorescence of the acceptor cell to reach 10% of that of the donor cell for the two conditions. Mann–Whitney test, *P* = 1.0, 35 *vs.* 55 mmHg. [Colour figure can be viewed at wileyonlinelibrary.com]

**Table 2. Cryogenic electron microscopy refinement and validation statistics**

| Refinement | GgCx26 |
|---|---|
| Initial model used (PDB code) | 7QEQ |
| Resolution (Å; FSC = 0.5) map | 2.2 |
| Sharpening B factor (Å$^2$) | 0 |
| **Model composition** | 12 subunits |
| Non-hydrogen atoms | 20,484 |
| Protein residues | 2256 |
| Water | 468 |
| Ligand: lipid/detergent | 48 |
| **B factor (Å$^2$) (min/max/mean)** | |
| Protein | 27/229/84 |
| Water | 46/134/74 |
| Lipid/detergent | 76/139/105 |
| **R.m.s. deviations** | |
| Bond lengths (Å) | 0.003 |
| Bond angles (°) | 0.490 |
| **Validation** | |
| MolProbity score | 1.24 |
| Clashscore | 3.43 |
| Rotamer outliers (%) | 1.38 |
| **Ramachandran plot** | |
| Favoured (%) | 0 |
| Allowed (%) | 1.81 |
| Disallowed (%) | 98.19 |
| **Map resolution (Å)** | |
| FSC threshold | 0.143 |
| Map resolution range (Å) (unmasked) | 2.0 |
| Map resolution range (Å) (masked) | 2.1 |
| **Correlation coefficients** | |
| CC (mask) | 0.93 |
| CC (box) | 0.74 |
| CC (peaks) | 0.73 |
| CC (volume) | 0.92 |
| Mean CC for ligands | 0.71 |

opened by $CO_2$. Some impact of these mutations on the $CO_2$ sensitivity may be expected because the introduction of the positively charged arginine residue into the cytoplasmic loop is likely to alter the chemical environment and, thus, the propensity for K125 to be carbamylated.

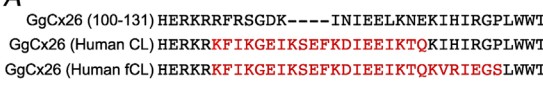

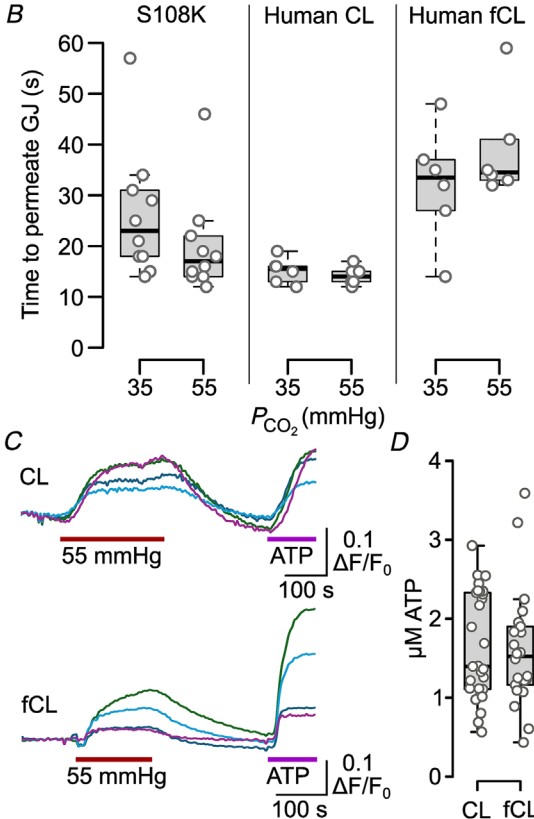

**Figure 10. Neither the mutation S108K nor insertion of the hCx26 cytoplasmic loop in GgCx26 restore gap junction channel sensitivity to $CO_2$**
*A*, sequence alignment of the intracellular loop of wild-type GgCx26 and the substituted human intracellular loops which lack (CL) or possess (fCL) the hCx26 carbamylation motif. *B*, neither the mutation S108K, nor insertion of the CL or fCL into GgCx26 rescues $CO_2$-dependent gap junction channel closure. The graphs show that the time for the fluorescence of the acceptor cell to reach 10% of that of the donor cell is insensitive to $CO_2$. Mann–Whitney test: $P = 0.16$ for S108K 35 *vs.* 55 mmHg; $P = 0.54$, human CL 35 *vs.* 55 mmHg; and $P = 0.47$ for human fCL 35 *vs.* 55 mmHg. *C*, insertion of the CL or fCL into GgCx26 does not disrupt the ability of $CO_2$ ($P_{CO_2}$ change from 20 to 55 mmHg) to open the chimeric hemichannels. Sample GRAB$_{ATP}$ recordings show responses to $P_{CO_2}$ of 55 mmHg (red bar) and 3 μM ATP calibration (magenta bar). Each coloured line represents changes in GRAB$_{ATP}$ fluorescence from a separate cell in the field of view. *D*, box plot shows $CO_2$-evoked ATP release from 29 cells from four independent transfections for CL, and 22 cells from three independent transfections for fCL. [Colour figure can be viewed at wileyonlinelibrary.com]

112, 116, 122 and 125. Whereas K112 and K116 are carbamylated at a $P_{CO_2}$ of 20 mmHg and thus would be expected to be constitutively modified under physiological conditions, K108, K122 and K125 require higher levels of $P_{CO_2}$ to become carbamylated. This substantiates a prediction from Huckstepp et al. (2010a) that multiple lysine residues could be carbamylated to mediate the actions of $CO_2$ on Cx26 and other beta connexins. We cannot rule out the possibility that other lysine residues not visible by MS/MS might also be carbamylated.

Although all five lysine residues in the intracellular loop can be modified by carbamylation, only carbamylation of K125 is essential for $CO_2$-dependent hemichannel opening (Meigh et al., 2013). Prevention of carbamylation of the other lysine residues by mutating them to arginine may have some effect on the $CO_2$ sensitivity of hemichannel opening. However, the hemichannel can still be

We have previously shown that the Cx26 gap junction channel is directly sensitive to $CO_2$ over the same concentration range as the hemichannel (Nijjar et al., 2021). However, the gap junction channel is closed rather than opened by $CO_2$. The action of $CO_2$ in closing the gap junction channel also depends on the carbamylation of K125 (Nijjar et al., 2021). Mimicking carbamylation of K125 by introducing a Glu residue at this position creates gap junction channels that are constitutively closed and insensitive to $CO_2$ (Brotherton et al., 2024). This same mutation creates hemichannels that are constitutively open (Meigh et al., 2013). The data we present here show that carbamylation of K108, in addition to K125, is required for $CO_2$-dependent gap junction channel closure but not for hemichannel opening.

Additional structural interactions and mechanisms are involved in gap junction closure compared with hemichannel opening to $CO_2$. Our data here show that $CO_2$-dependent hemichannel opening can be dissociated from gap junction closure even though they still depend on the same Lys residue (K125). We have previously reported a dissociation between $CO_2$-dependent gap junction channel closure and hemichannel opening. In fish and amphibian homologues of Cx26, an extended C-terminus prevents $CO_2$-dependent hemichannel opening but still permits $CO_2$-dependent gap junction channel closure (Dospinescu et al., 2019). Presumably, the carbamylated K108 interacts with other residues to enable gap junction closure. These interactions either do not occur in the hemichannel, which will have a somewhat different conformation from the gap junction or are simply not required for hemichannel opening to $CO_2$. One possibility for interaction is Arg216 in TM4. Mutation of this residue has a similar phenotype to mutating Lys108, in that both abolish gap junction closure by $CO_2$ but, crucially, do not prevent hemichannel opening to $CO_2$. The further observation that gap junction channels with the double mutation K108R-R216E are constitutively closed is consistent with this hypothesis. However, the latest structure of Cx26 gap junction channels in the closed state does not provide evidence for a close interaction between K108 and R216

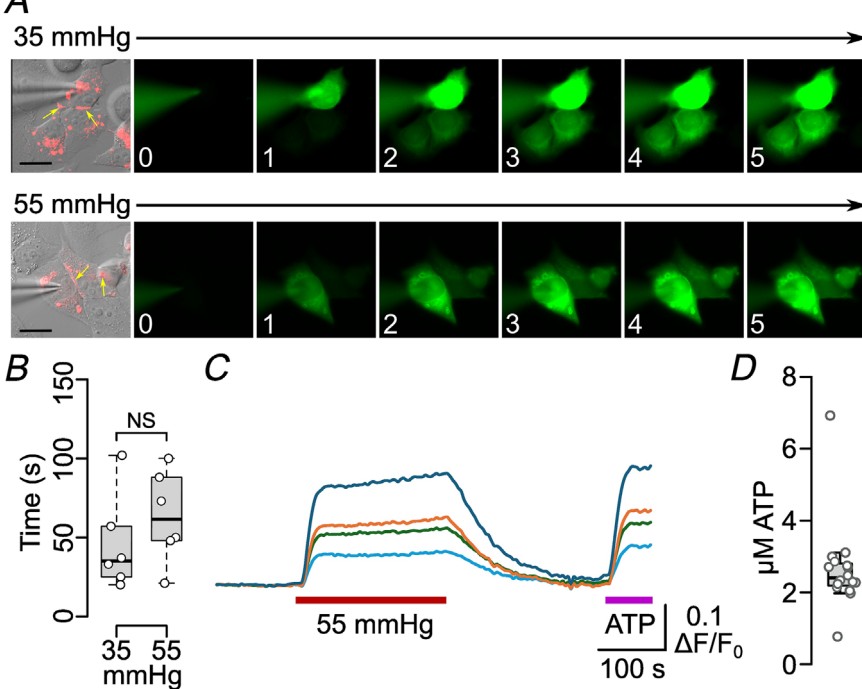

**Figure 11. R216Q abolishes $CO_2$-dependent gap junction channel closure**
*A*, hCx26[R216Q] tagged with mCherry was expressed in HeLa cells. Gap junctions are evident as a red stripe (yellow arrows). The permeation of dye from donor to acceptor cell was equally fast at 35 and 55 mmHg $P_{CO_2}$. In both examples, dye permeates to a third cell that is also coupled via a gap junction. Numbers in bottom left corners of the fluorescence images indicate minutes after establishment of whole-cell recording. Scale bars 20 μm. *B*, summary data for the time for the fluorescence in the acceptor cell to reach 10% of the fluorescence of the donor cell. There is no difference between the time to permeate at 35 and 55 mmHg (Mann–Whitney test *P* = 0.394, 35 *vs.* 55 mmHg). *C*, hCx216[R216Q] hemichannels respond to $CO_2$ (20 to 55 mmHg $P_{CO_2}$, red bar). GRAB$_{ATP}$ fluorescence changes evoked in four cells by a change $P_{CO_2}$, compared with change evoked by 3 μM ATP calibration. *D*, box plot shows the concentration change for $CO_2$-evoked ATP release evoked by 55 mmHg $P_{CO_2}$ for 16 cells from three independent transfections. [Colour figure can be viewed at wileyonlinelibrary.com]

(Brotherton et al., 2024). Therefore, the precise role of R216 remains uncertain. For example, R216 may only be indirectly involved or there may be additional potential interacting residues in TM4 and the C-terminus that remain to be tested.

The initial clue that GgCx26 gap junction channels might not close to $CO_2$ came from our cryo-EM studies. For hCx26, the closed conformation of the gap junction channel involves the N-termini being ordered within the pore to form a constriction at the centre (Brotherton et al., 2022). Our cryo-EM reconstructions of GgCx26 vitrified at a $P_{CO_2}$ of 90 mmHg do not show this characteristic ordering of the N-termini and appear much more like the open structure of hCx26 at low $P_{CO_2}$. The exact mechanism of channel closure in connexin gap junction channels remains controversial. Based on the structures of Cx46/50 (Flores et al., 2020; Myers et al., 2018), Cx43 (Lee et al., 2023a; Qi et al., 2023) and Cx36 (Lee et al., 2023b), the N-termini have been proposed to be located in a downward position tucked back against the pore, whereas in the more closed state the N-termini are raised to form a cap outside the channel pore with lipids important in restricting entrance to the channel. While we observe a lipid molecule within the pore, it does not appear to be important in the regulatory mechanism. Although we are comparing Cx26 from different species, our data further strengthen the idea that the ordered alignment of N-termini within the channel pore is involved in closing the Cx26 channel.

Carbamylation, also known as carboxylation, is a spontaneous non-enzymatic post-translational modification of lysine residues via a labile covalent bond between the N of a primary amine and the C of $CO_2$. Given that five lysine residues in the cytoplasmic loop of hCx26 become carbamylated, our data show that this loop must provide a chemical microenvironment in which the pKa of all of these lysine side chains is lowered. The structural biology that explains why the cytoplasmic loop provides this microenvironment and whether interaction with the C-terminus may be important remains unknown because this region is highly flexible (Bennett et al., 2016; Brotherton et al., 2022; Maeda et al., 2009). Although carbamylation lacks the specificity of enzymatic post-translational modifications such as phosphorylation or acetylation, it can nevertheless impart specific changes in protein function. As the specificity of carbamylation is encoded within the structure of the protein itself, carbamylation could be considered a very primitive post-translational modification of a protein, as it only requires the presence of $CO_2$ at a sufficient concentration. In the context of Cx26, we have been able to assign specific and differential modulatory actions on the gap junction and hemichannel for two of the five carbamylated lysine residues. For Cx26, this $CO_2$ sensitivity is an important

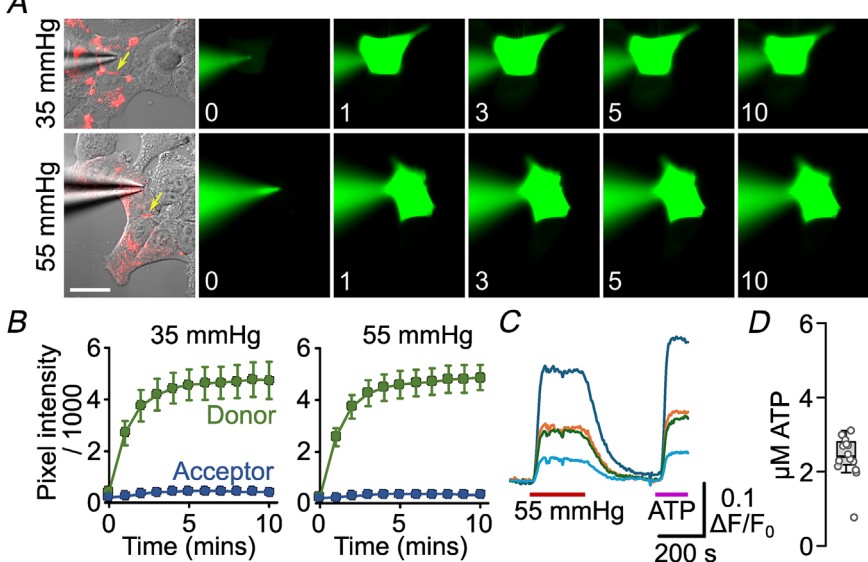

**Figure 12. K108R-R216E gap junction channels are shut at high and low $P_{CO_2}$**
*A*, hCx26[K108R,R216E] tagged with mCherry was expressed in HeLa cells. Gap junctions are evident as a red stripe (yellow arrows). No dye permeation donor to acceptor cell was observed at either 35 or 55 mmHg $P_{CO_2}$. Numbers in bottom left corners of the fluorescence images indicate minutes after establishment of whole-cell recording. Scale bar 20 µm. *B*, summary time course data (means ± SEM) for the time course of the fluorescence changes in the donor and acceptor cells at 35 and 55 mmHg (*n* = 6 cells for each level of $P_{CO_2}$). *C*, hCx26[K108R,R216E] hemichannels respond to $CO_2$ (20–55 mmHg $P_{CO_2}$, red bar). GRAB_ATP fluorescence changes evoked by $CO_2$ and 3 µM ATP in four cells. *D*, box plot shows plot of the change evoked in ATP concentration evoked by 55 mmHg $P_{CO_2}$ for 22 cells from three independent transfections. [Colour figure can be viewed at wileyonlinelibrary.com]

contributor to the chemosensory control of breathing, showing that carbamylation is not just a biochemical epiphenomenon but plays an essential physiological role (van de Wiel et al., 2020). As the carbamylation motif has been conserved within the beta connexins of a range of vertebrate species (specifically Cx26, Cx26-like, 30 and 32) over an evolutionary distance of more than 400 MY (Dospinescu et al., 2019), this implies that this mechanism of connexin modulation must have a series of important roles across vertebrates as diverse as sharks and humans; otherwise, it would have been lost by random genetic drift.

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

## Additional information

### Data availability statement

The cryo-EM density maps have been deposited in the Electron Microscopy Data Bank under accession numbers EMD-18670 and the associated PDB in the RCSB Protein Data Bank with accession number 8QVN. All other data are available in the supplementary information.

### Competing interests

The authors declare that they have no competing interests.

## Author contributions

S.N., D.B., J.B., V.M.D., H.G.G., V.L.: data collection and analysis. M.C., A.C., N.D.: writing, study design, analysis. All authors: approved the final version of the manuscript; agree to be accountable for all aspects of the work in ensuring that questions related to the accuracy or integrity of any part of the work are appropriately investigated and resolved; and all persons designated as authors qualify for authorship, and all those who qualify for authorship are listed.

## Funding

We thank the Leverhulme Trust (RPG-2015-090, ND; F/00/128/AU, MJC), MRC (MR/P010393/1, ND) and BBSRC (BB/T013346/1, ND; BB/S015132/1, MJC) for support. JB was supported by the Biotechnology and Biological Sciences Research Council (BBSRC) and University of Warwick-funded Midlands Integrative Biosciences Training Partnership (MIBTP) grant number BB/T00746X/1. VMD was funded by the Medical Research Council through the University of Warwick Doctoral Training Partnership, grant number MR/N014294/1.

## Acknowledgements

The authors acknowledge the Midlands Regional Cryo-EM Facility, hosted at the Warwick Advanced Bioimaging Research Technology Platform, for use of the JEOL 2100Plus, and the Midlands Regional Cryo-EM Facility, hosted at Leicester Institute of Structural and Chemical Biology for use of the FEI Titan Krios G3, both supported by MRC award reference MC_PC_17136. The authors are grateful to Dr Christos Savva, TJ Ragan in Leicester for help with image collection and processing and the technical support in the School of Life Sciences, University of Warwick.

## Keywords

carbamylation, carboxylation, connexin, cryo-EM, gap junction channel, hemichannel

## Supporting information

Additional supporting information can be found online in the Supporting Information section at the end of the HTML view of the article. Supporting information files available:

**Peer Review History**
**Supporting information**
**Supporting information**
**Supporting information**
**Supporting information**
**Supplementary information**
**Supporting information**
**Supporting information**
**Supporting information**

