## [Peer Review History · The Journal of Physiology]

Multiple carbamylation events are required for differential modulation of Cx26 hemichannels and gap junctions by CO₂

Sarbjit Nijjar, Deborah Brotherton, Jack Butler, Valentin-Mihai Dospinescu, Harry Gannon, Victoria Linthwaite, Martin J. Cann, Alexander Cameron, and Nicholas Dale
DOI: 10.1113/JP285885

Corresponding author(s): Nicholas Dale (N.E.Dale@warwick.ac.uk)

The following individual(s) involved in review of this submission have agreed to reveal their identity: Agustin Demetrio Martinez (Referee #1)

Review Timeline:

Submission Date:	31-Oct-2023
Editorial Decision:	05-Dec-2023
Revision Received:	28-Nov-2024
Accepted:	13-Jan-2025

Senior Editor: Kim Barrett

Reviewing Editor: Rajini Rao

Transaction Report:

Dear Dr Dale,

Re: JP-RP-2023-285885 "Multiple carbamylation events are required for differential modulation of Cx26 hemichannels and gap junctions by CO₂" by Sarbjit Nijjar, Deborah Brotherton, Valentin-Mihai Dospinescu, Harry Gannon, Victoria Linthwaite, Martin J. Cann, Alexander cameron, and Nicholas Dale

Thank you for submitting your manuscript to The Journal of Physiology. It has been assessed by a Reviewing Editor and by 2 expert referees and we are pleased to tell you that it is potentially acceptable for publication following satisfactory major revision.

LANGUAGE EDITING AND SUPPORT FOR PUBLICATION: If you would like help with English language editing, or other article preparation support, Wiley Editing Services offers expert help, including English Language Editing, as well as translation, manuscript formatting, and figure formatting at www.wileyauthors.com/eoo/preparation. You can also find resources for Preparing Your Article for general guidance about writing and preparing your manuscript at www.wileyauthors.com/eoo/prepresources.

REVISION CHECKLIST:

Please upload two versions of your manuscript text: one with all relevant changes highlighted and one clean version with no changes tracked. The manuscript file should include all tables and figure legends, but each figure/graph should be uploaded as separate, high-resolution files. The journal is now integrated with Wiley's Image Checking service. For further details, see: <https://www.wiley.com/en-us/network/publishing/research-publishing/trending-stories/upholding-image-integrity-wileys->

image-screening-service

We look forward to receiving your revised submission.

Yours sincerely,

Professor Kim E. Barrett
Editor-in-Chief
The Journal of Physiology
<https://jp.msubmit.net>
<http://jp.physoc.org>
The Physiological Society
Hodgkin Huxley House
30 Farringdon Lane
London, EC1R 3AW
UK
<http://www.physoc.org>
<http://journals.physoc.org>

REQUIRED ITEMS

- Author photo and profile. First or joint first authors are asked to provide a short biography (no more than 100 words for one author or 150 words in total for joint first authors) and a portrait photograph. These should be uploaded and clearly labelled together in a Word document with the revised version of the manuscript. See Information for Authors for further details.
- You must start the Methods section with a paragraph headed Ethical Approval. If experiments were conducted on humans, confirmation that informed consent was obtained, preferably in writing, that the studies conformed to the standards set by the latest revision of the Declaration of Helsinki and that the procedures were approved by a properly constituted ethics committee, which should be named, must be included in the article file. If the research study was registered (clause 35 of the Declaration of Helsinki), the registration database should be indicated, otherwise the lack of registration should be noted as an exception (e.g. The study conformed to the standards set by the Declaration of Helsinki, except for registration in a database). For further information see: <https://physoc.onlinelibrary.wiley.com/hub/human-experiments>.
- The reference list must be in alphabetical order, rather than numbered, to comply with our Journal format.
- Your manuscript must include a complete Additional Information section, including competing interests; funding; author contributions and acknowledgements.
- The Journal of Physiology funds authors of provisionally accepted papers to use the premium BioRender site to create high resolution schematic figures. Follow this link and enter your details and the manuscript number to create and download figures. Upload these as the figure files for your revised submission. If you choose not to take up this offer, we require figures to be of similar quality and resolution. If you are opting out of this service to authors, state this in the Comments section on the Detailed Information page of the submission form. The link provided should only be used for the purposes of this submission. Authors will be charged for figures created on this premium BioRender account if they are not related to this manuscript submission.
- Please upload separate high-quality figure files via the submission form.
- Please ensure that any tables are editable and in Word format, and wherever possible, embedded in the article file itself.
- Please ensure that the Article File you upload is a Word file.
- Papers must comply with the Statistics Policy: https://jp.msubmit.net/cgi-bin/main.plex?form_type=display_requirements#statistics.

In summary:

- If n {less than or equal to} 30, all data points must be plotted in the figure in a way that reveals their range and distribution.

A bar graph with data points overlaid, a box and whisker plot or a violin plot (preferably with data points included) are acceptable formats.

- If $n > 30$, then the entire raw dataset must be made available either as supporting information, or hosted on a not-for-profit repository, e.g. FigShare, with access details provided in the manuscript.

- 'n' clearly defined (e.g. x cells from y slices in z animals) in the Methods. Authors should be mindful of pseudoreplication.

- All relevant 'n' values must be clearly stated in the main text, figures and tables.

- The most appropriate summary statistic (e.g. mean or median and standard deviation) must be used. Standard Error of the Mean (SEM) alone is not permitted.

- Exact p values must be stated. Authors must not use 'greater than' or 'less than'. Exact p values must be stated to three significant figures even when 'no statistical significance' is claimed.

- Please include an Abstract Figure file, as well as the Figure Legend text within the main article file. The Abstract Figure is a piece of artwork designed to give readers an immediate understanding of the research and should summarise the main conclusions. If possible, the image should be easily 'readable' from left to right or top to bottom. It should show the physiological relevance of the manuscript so readers can assess the importance and content of its findings. Abstract Figures should not merely recapitulate other figures in the manuscript. Please try to keep the diagram as simple as possible and without superfluous information that may distract from the main conclusion(s). Abstract Figures must be provided by authors no later than the revised manuscript stage and should be uploaded as a separate file during online submission labelled as File Type 'Abstract Figure'. Please also ensure that you include the figure legend in the main article file. All Abstract Figures should be created using BioRender. Authors should use The Journal's premium BioRender account to export high-resolution images. Details on how to use and access the premium account are included as part of this email.

- Please include a full title page as part of your main article (Word) file, which should contain the following: title, authors, affiliations, corresponding author name and contact details, keywords, and running title.

- Please ensure that all figures and tables have a title and legend, and that they have been cited within the main article text.

EDITOR COMMENTS

Reviewing Editor:

Your paper has been evaluated by two expert reviewers who find the work to be potentially quite influential to the field of cell physiologists. Reviewer 1 recommends additional control experiments, in particular, quantification of mutant expression at the plasma membrane. Reviewer 2 suggests minor edits to the text that do not require additional experiments. We believe that your study should be re-evaluated following revision as recommended.

REFEREE COMMENTS

Referee #1:

The article entitle "Multiple carbamylation events are required for differential modulation of Cx26 hemichannels and gap junctions by CO₂" from authors Nijjar et al., shows that multiple lysine residues within the IL of hCx26 can be carbamylated at physiologic PCO₂. However, only mutation in Lys125 impedes the opening of Hemichannels (HCs) and the closing of Gap Junction channels (GJC) induced by CO₂. Moreover, mutation in Lys108 also affects the closure of GJC induced by CO₂. The authors also present data from Cryo-EM molecular structure of Chicken Cx26 GJs that lack Lys108 obtained at 15% CO₂ showing a structure that resemble the open GJC, supporting that Lys108 is also critic for GJC regulation induced by CO₂. Finally, they provide evidence that a possible salt bridge between Lys108 and R206 is necessary for GJC closure induced by CO₂.

This article provides new information on the possible regulation of these channels by CO₂ and will undoubtedly contribute to generating a better understanding of the functioning of these channels. Yet, this study lacks some important information to fully support some of the interpretations established by the authors to support the explanations they propose.

1.- The experiments with the lysine mutants require more characterization with respect to the effect of respective mutation on Hemichannels and Gap Junction levels at the plasma membrane. For examples, in FIG2 A it seems that at least mutants K108R, K112R and K122R present less Gap Junction plaques and more expression of these mutants in cell cytoplasm. Therefore, I think it is necessary to show the levels of Cx26 at plasma membrane, for example, with protein biotinylation or with a confocal analysis and some type of quantification to better established that the conditions could be comparable.

It is known that other posttranslational modifications in Cxs targeting Lysine, like Ubiquitination, and that this type of modification may affect the amount and localization of connexin channels at plasma membrane. Moreover, less gap junction plaques could also indicate more free hemichannels at plasma membrane. This possibility could explain some of the large variability observed with some mutants, for example, in figure 2B mutant K108R, which presents some experiments with large uptakes and others with reduced uptake.

In addition, the possible changes in the initial amount of channels at plasma membrane could also explain the differences observed in figure 4 and 5, where K122R has large whole cell currents or conductance induced by low levels of CO₂ compared to the double mutant K108R, K122R or K125R.

2.- In addition, provides evidence that the dye uptake or whole cell current occurs through hemichannels by incorporating known hemichannel blockers like carbenoxolone or La³⁺ or other in some of your experiments, as a necessary control. This is because HeLa cells present low levels of Cx45 (very few) but express pannexin channels, especially Panx1.

3.- The finding with Cryo-EM structure of the CgCx26 in high PCO₂ presents an open pore, which is very interesting and supports a possible mechanism for the regulation by PCO₂ on GJC. All these experiments are quite clear and show the participation of Lys108 and R206, possibly by the formation of a salt bridge between both residues. Although the experiment with the mutation R206Q shows the participation of R206 in the sensitivity to CO₂, the formation of a salt bridge could be better demonstrated with the double mutant that could recover the salt bridge formation, Lys108R, R206Lys, that could be nice to have.

Referee #2:

This manuscript presents a straightforward demonstration of the role of carbamylation of other Lysine residues not tested previously on the activity of Cx26 hemichannels and gap junction channels. The data is clearly presented and was obtained with appropriate techniques.

Some minor issues need to be clarified throughout the manuscript.

Introduction. Third paragraph, first line. Cx26, like all connexins...not all Cxs form hemichannels and gap junction channels (please see <https://doi.org/10.3389/fphys.2017.00038>).

Same paragraph, 4th line. A gap junction channel...(A gap junction is a cluster of a variable number of gap junction channels). Please fix this throughout the manuscript.

Last paragraph. ...within the cytoplasmic loop are capable of carbamylation

Methods. CO₂ trapping. Purified hCx26 (0.7 mg) in the stabilizing buffer...this is not a buffer, please change it by the solution?

Results,

Since Cx26 hemichannels open and close under resting conditions (doi: 10.1096/fj.06-5828com), I recommend describing that CO₂ increases the activity of hemichannels and reduces the activity of gap junction channels open probability. It might be more accurate to say increase in Cx26 hemichannel activity.

Please change the dose by concentration. Dose is referred to gr/kg of body weight.

As for other connexin structures no density was apparent for the cytoplasmic loop between residues Arg 105 and

Arg 125 (equivalent to Lys105 and Glu129 of hCx26 respectively). This sentence requires a couple of references.

END OF COMMENTS

REQUIRED ITEMS

- Author photo and profile. First or joint first authors are asked to provide a short biography (no more than 100 words for one author or 150 words in total for joint first authors) and a portrait photograph. These should be uploaded and clearly labelled together in a Word document with the revised version of the manuscript. See Information for Authors for further details.

Provided

- You must start the Methods section with a paragraph headed Ethical Approval. If experiments were conducted on humans, confirmation that informed consent was obtained, preferably in writing, that the studies conformed to the standards set by the latest revision of the Declaration of Helsinki and that the procedures were approved by a properly constituted ethics committee, which should be named, must be included in the article file. If the research study was registered (clause 35 of the Declaration of Helsinki), the registration database should be indicated, otherwise the lack of registration should be noted as an exception (e.g. The study conformed to the standards set by the Declaration of Helsinki, except for registration in a database). For further information see: <https://physoc.onlinelibrary.wiley.com/hub/human-experiments>.

No animal or human experiments reported

- The reference list must be in alphabetical order, rather than numbered, to comply with our Journal format.

Done

- Your manuscript must include a complete Additional Information section, including competing interests; funding; author contributions and acknowledgements.

Done

- The Journal of Physiology funds authors of provisionally accepted papers to use the premium BioRender site to create high resolution schematic figures. Follow this link and enter your details and the manuscript number to create and download figures. Upload these as the figure files for your revised submission. If you choose not to take up this offer, we require figures to be of similar quality and resolution. If you are opting out of this service to authors, state this in the Comments section on the Detailed Information page of the submission form. The link provided should only be used for the purposes of this submission. Authors will be charged for figures created on this premium BioRender account if they are not related to this manuscript submission.

- Please upload separate high-quality figure files via the submission form.

- Please ensure that any tables are editable and in Word format, and wherever possible, embedded in the article file itself.

- Please ensure that the Article File you upload is a Word file.

- Papers must comply with the Statistics Policy: https://jp.msubmit.net/cgi-bin/main.plex?form_type=display_requirements#statistics.

In summary:

- If $n \leq 30$, all data points must be plotted in the figure in a way that reveals their range and distribution. A bar graph with data points overlaid, a box and whisker plot or a violin plot (preferably with data points included) are acceptable formats.

- If $n > 30$, then the entire raw dataset must be made available either as supporting information, or hosted on a not-for-profit repository, e.g. FigShare, with access details provided in the manuscript.

- 'n' clearly defined (e.g. x cells from y slices in z animals) in the Methods. Authors should be mindful of pseudoreplication.

- All relevant 'n' values must be clearly stated in the main text, figures and tables.

- The most appropriate summary statistic (e.g. mean or median and standard deviation) must be used. Standard Error of the Mean (SEM) alone is not permitted.

- Exact p values must be stated. Authors must not use 'greater than' or 'less than'. Exact p values must be stated to three significant figures even when 'no statistical significance' is claimed.

- Please include an Abstract Figure file, as well as the Figure Legend text within the main article file. The Abstract Figure is a piece of artwork designed to give readers an immediate understanding of the research and should summarise the main conclusions. If possible, the image should be easily 'readable' from left to right or top to bottom. It should show the physiological relevance of the manuscript so readers can assess the importance and content of its findings. Abstract Figures should not merely recapitulate other figures in the manuscript. Please try to keep the diagram as simple as possible and without superfluous information that may distract from the main conclusion(s). Abstract Figures must be provided by authors no later than the revised manuscript stage and should be uploaded as a separate file during online submission labelled as File Type 'Abstract Figure'. Please also ensure that you include the figure legend in the main article file. All Abstract Figures should be created using BioRender. Authors should use The Journal's premium BioRender account to export high-resolution images. Details on how to use and access the premium account are included as part of this email.

Done

- Please include a full title page as part of your main article (Word) file, which should contain the following: title, authors, affiliations, corresponding author name and contact details, keywords, and running title.

Done

- Please ensure that all figures and tables have a title and legend, and that they have been cited within the main article text.

Done

EDITOR COMMENTS

Reviewing Editor:

Your paper has been evaluated by two expert reviewers who find the work to be potentially quite influential to the field of cell physiologists. Reviewer 1 recommends additional control experiments, in particular, quantification of mutant expression at the plasma membrane. Reviewer 2 suggests minor edits to the text that do not require additional experiments. We believe that your study should be re-evaluated following revision as recommended.

REFEREE COMMENTS

Referee #1:

The article entitle "Multiple carbamylation events are required for differential modulation of Cx26 hemichannels and gap junctions by CO₂" from authors Nijjar et al., shows that multiple lysine residues within the IL of hCx26 can be carbamylated at physiologic PCO₂. However, only mutation in Lys125 impedes the opening of Hemichannels (HCs) and the closing of Gap Junction channels (GJC) induced by CO₂. Moreover, mutation in Lys108 also affects the closure of GJC induced by CO₂. The authors also present data from Cryo-EM molecular structure of Chicken Cx26 GJs that lack Lys108 obtained at 15% CO₂ showing a structure that resemble the open GJC, supporting that Lys108 is also critic for GJC regulation induced by CO₂. Finally, they provide evidence that a possible salt bridge between Lys108 and R206 is necessary for GJC closure induced by CO₂.

This article provides new information on the possible regulation of these channels by CO₂ and will undoubtedly contribute to generating a better understanding of the functioning of these channels. Yet, this study lacks some important information to fully support some of the interpretations established by the authors to support the explanations they propose.

1.- The experiments with the lysine mutants require more characterization with respect to the effect of respective mutation on Hemichannels and Gap Junction levels at the plasma membrane. For examples, in FIG2 A it seems that at least mutants K108R, K112R and K122R

present less Gap Junction plaques and more expression of these mutants in cell cytoplasm. Therefore, I think it is necessary to show the levels of Cx26 at plasma membrane, for example, with protein biotinylation or with a confocal analysis and some type of quantification to better established that the conditions could be comparable.

We have performed membrane staining with DiO on fixed cells (to avoid membrane turnover and internalisation of DiO) and quantified its colocalization to the mCherry tag on the Cx26 variants. We show the Mander's coefficient for mCherry to DiO localisation. This is the same in all mutants apart from K122R which shows significantly lower amounts of membrane localisation. This observation about the altered expression of K122R was highlighted in the first version of our MS. We note that the lower colocalization of K122R mutants in the plasma membrane did not correlate with lower sensitivity of the whole cell currents, dye loading and ATP release to CO₂. However, with this mutant, it was much harder to find suitable cells for analysis of these measures due to this altered pattern of expression.

It is know that other posttranslational modification in Cxs targeting Lysine, like Ubiquitination, and that this type of modification may affect the amount and localization of connexins channels at plasma membrane. Moreover, less gap junction plaques could also indicates more free hemichannels at plasma membrane. This possibility could explain some of the large variability observed with some mutants, for example, in figure 2B mutant K108R, which present some experiments with large uptakes and other with reduced uptake.

The large variability in the dye loading results, we suspect, was due to the relative inexperience of the experimenter at the time. This method, although conceptually simple, requires a certain level of expertise to perform reliably and we have a strict policy of showing all data no matter how imperfect.

We have sought to give further reassurance on the hemichannel data by providing the results of a new assay that we have recently developed (Butler & Dale 2023) -real time measurement of ATP release via coexpression of GRAB_{ATP}, a genetically encoded ATP sensor. This shows that HeLa cells expressing all mutants, apart from K125R exhibit CO₂ dependent ATP release, and that HeLa cells that do not express Cx26 do not show this ATP release. These new data are presented in Figure 4.

In addition, the possible changes in the initial amount of channels at plasma membrane could also explain the differences observed in figure 4 and 5, where K122R have large whole cell currents or conductance induced by low levels of CO₂ compared to the double mutant K108R, K122R or K125R.

As pointed out above K122R has poorer expression at the plasma membrane, not better, so that this explanation cannot be the case. Equally all of the other mutants express to similar levels.

We have documented the dose sensitivity of the mutant channels to CO₂, in addition to the absolute sensitivity at a particular level of CO₂. It is clear that all mutants apart from K125R, which completely removes CO₂ sensitivity, shift dose dependence to lower levels of CO₂ (dye loading and ATP release). This cannot be explained by alterations of levels of expression, as

the dose dependence measures the affinity of the channels to CO₂ rather than their density in the membrane. Nevertheless, the additional colocalization analysis with a membrane marker shows that the pattern of expression of the different mutants is highly similar with the exception of K122R.

2.- In addition, provides evidences that the dye uptake or whole cell current occurs through hemichannels by incorporating known hemichannels blockers like carbenoxolone or La³⁺ or other in some of your experiments, as a necessary control. This is because HeLa cells present low levels of Cx45 (very few) but express pannexin channels, especially Panx1.

We have provided these controls or equivalents in many of our prior studies, but to address this we now document with new data that parental HeLa cells used in this study show only a minimal change in whole cell conductance to changes in PCO₂ from 35 to 55 mmHg. We also demonstrate that application of 200 μM La³⁺ completely blocks the CO₂ dependent changes in whole cell conductance in Cx26-expressing HeLa cells Cx26 (Figure 5). We note also that the GRAB-only control shows the requirement for connexin expression for CO₂ dependent ATP release (Figure 4).

For completeness, we note that when our current parental HeLa DH cells (from ECACC) are placed in aCSF at 20 mmHg and then transferred to 55 mmHg or higher, an outward current and conductance change can be observed in some cells. We have not determined the mechanistic basis for this but note that we do not observe this conductance change in parental HeLa Ohio cells (or in the original parental cells used in Huckstepp et al 2010 (Figure 2)). As a matter of caution therefore, we have removed from the paper any patch clamp recordings that used 20 mmHg aCSF as a starting point. We have substituted GRAB_{ATP} recordings as an assay of hemichannel gating from 20 mmHg where necessary (Figs 10, 11 and 12), as there is no change in GRAB_{ATP} fluorescence in parental HeLa cells to changes in PCO₂ from a starting point of 20 mmHg (Figure 4).

3.- The finding with Cryo-EM structure of the CgCx26 in high PCO₂ present a open pore is very interesting and support a possible mechanism for the regulation by PCO₂ on GJC. All these experiments are quite clear and show the participation of Lys108 and R206, possible by the formation of a salt bridge between both residues. Although the experiment with the mutation R206Q show the participation of R206 in the sensitivity to CO₂, the formation a salt bridge could be better demonstrate with the double mutant that could recover the salt bridge formation, Lys108R, R206Lys, that could be nice to have.

We agree with these comments (we think you mean R216Q for hCx26). We did indeed perform some additional experiments to test the salt bridge hypothesis prior to submission of the original MS, namely we evaluated the GgCx26 S108K-K212R, to see if this gave a gain of CO₂ sensitivity to this channel. Here, K212 is the equivalent of R216 in hCx26. We got quite excited -the gap junction was closed at a PCO₂ of 35 mmHg, but unfortunately it was also closed when we completely removed CO₂ by using a HEPES based buffering system. We concluded that this was not a gain of CO₂ sensitivity in this double mutant. We decided not to include this data as it provided no further insight and could be potentially misleading.

However, stimulated by the referee's thoughts we have now assayed the double mutant K108R-R216E in hCx26. We think this is preferable to the referee's suggestion of K108R-R216K, as there is no guarantee that a Lys residue in position 216 would become carbamylated. The R216E mutation places a negative charge on R216 and could thus to form a "reverse" bridge with the now positively charged K108R. We also note that this strategy worked in the original Meigh at al 2013 paper, where R104E gave a constitutively open hemichannel. This mutation gives a constitutively closed gap junction channels at low PCO₂, so supports the hypothesis of a salt bridge between these residues. We used GRAB_{ATP} imaging to show that the hemichannel remains sensitive to CO₂. We now document this in Figure 12. We note that our most recent structure for the Cx26 GJC (eLife DOI: 10.7554/eLife.93686P) does not support a direct interaction between K108 and R216. We raise this point in the Discussion along with appropriate caveats in interpreting the mutational data.

Referee #2:

This manuscript presents a straightforward demonstration of the role of carbamylation of other Lysine residues not tested previously on the activity of Cx26 hemichannels and gap junction channels. The data is clearly presented and was obtained with appropriate techniques.

Some minor issues need to be clarified throughout the manuscript.

Introduction. Third paragraph, first line. Cx26, like all connexins...not all Cxs form hemichannels and gap junction channels (please see <https://doi.org/10.3389/fphys.2017.00038>).

Thanks for this -we have adjusted the wording to avoid this implication

Same paragraph, 4th line. A gap junction channel...(A gap junction is a cluster of a variable number of gap junction channels). Please fix this throughout the manuscript.

We have done this

Last paragraph. ...within the cytoplasmic loop are capable of carbamylation

Changed

Methods. CO₂ trapping. Purified hCx26 (0.7 mg) in the stabilizing buffer...this is not a buffer, please change it by the solution?

Changed as suggested

Results,

Since Cx26 hemichannels open and close under resting conditions (doi: 10.1096/fj.06-5828com), I recommend describing that CO₂ increases the activity of hemichannels and reduces the activity of gap junction channels open probability. It might be more accurate to say increase in Cx26 hemichannel activity.

Yes we agree, indeed our own evidence shows that Cx26 is partially open at resting physiological PCO₂ (Huckstepp et al 2010 papers). Nevertheless, reduction of PCO₂ to 20 mmHg greatly reduces hemichannel opening -probably completely, although this is hard to be sure without completely removing CO₂. We would argue that the only two physiological ways to open Cx26 are an increase in PCO₂, or a very large increase in transmembrane voltage. The latter is not relevant here since HeLa cells are not excitable, and the patch clamp recordings were performed under voltage clamp at -50 mV. More generally as Cx26 is usually expressed in non-excitabile cells, it is hard to envisage transmembrane voltage as a physiological gating parameter.

Please change the dose by concentration Dose is referred to gr/kg of body weight.

This has been done.

As for other connexin structures no density was apparent for the cytoplasmic loop between residues Arg 105 and Arg 125 (equivalent to Lys105 and Glu129 of hCx26 respectively). This sentence requires a couple of references.

We agree and have added some references.

Dear Professor Dale,

Re: JP-RP-2024-285885R1 "Multiple carbamylation events are required for differential modulation of Cx26 hemichannels and gap junctions by CO₂" by Sarbjit Nijjar, Deborah Brotherton, Jack Butler, Valentin-Mihai Dospinescu, Harry Gannon, Victoria Linthwaite, Martin J. Cann, Alexander cameron, and Nicholas Dale

We are pleased to tell you that your paper has been accepted for publication in The Journal of Physiology.

Yours sincerely,

Kim Barrett
Senior Editor
The Journal of Physiology

If you would like to receive our 'Research Roundup', a monthly newsletter highlighting the cutting-edge research published in The Physiological Society's family of journals (The Journal of Physiology, Experimental Physiology, Physiological Reports, The Journal of Nutritional Physiology and The Journal of Precision Medicine: Health and Disease), please click this link, fill in your name and email address and select 'Research Roundup':

<https://www.physoc.org/journals-and-media/membernews>

- You can help your research get the attention it deserves! Check out Wiley's free Promotion Guide for best-practice recommendations for promoting your work at: www.wileyauthors.com/eo/guide. You can learn more about Wiley Editing Services which offers professional video, design, and writing services to create shareable video abstracts, infographics, conference posters, lay summaries, and research news stories for your research at: www.wileyauthors.com/eo/promotion.

EDITOR COMMENTS

Reviewing Editor:

We appreciate the additional experiments and clarifications in the revision and believe that this contribution is a significant advance to the field. Congratulations!

REFeree COMMENTS

Referee #2:

This manuscript is a logical continuation of previous reports. It is a robust manuscript with a good chance to make a relevant and positive impact in the field.

Thank you very much for the new data and for fixing the manuscript according to the reviewer's comments.